



# Lagged effects dominate the inter-annual variability of the 2010-2015 tropical carbon balance

A. Anthony Bloom[1], Kevin W. Bowman[1], Junjie Liu[1], Alexandra G. Konings[2], John R. Worden[1], Nicholas C, Parazoo[1], Victoria Meyer[1], John T. Reager[1], Helen M. Worden[5], Zhe Jiang[6], Gregory R. Quetin[2], T. Luke Smallman[3,4], Jean-François Exbrayat[3,4], Yi Yin[1], Sassan S. Saatchi[1], Mathew Williams[3,4], David S. Schimel[1].

[1]Jet Propulsion Laboratory, California Institute of Technology, Pasadena, CA 91101, U.S.A.
[2]Department of Earth System Science, Stanford University, Stanford, CA 94305, U.S.A.
[3]School of Geosciences, University of Edinburgh, Edinburgh, EH9 3FF, United Kingdom.
[4]National Centre for Earth Observation, Edinburgh EH9 3FF, United Kingdom.
[5]National Center for Atmospheric Research, Boulder, 80301 CO, U.S.A.
[6] School of Earth and Space Sciences, University of Science and Technology of China, Hefei, 230026, China.

*Correspondence to*: A. Anthony Bloom (abloom@jpl.nasa.gov)

**Abstract.**

Inter-annual variations in the tropical land carbon (C) balance are a dominant component of the global atmospheric $CO_2$ growth rate. Currently, the lack of quantitative knowledge on processes controlling net tropical ecosystems C balance on inter-annual timescales inhibits accurate understanding and projections of land-atmosphere C exchanges. In particular, uncertainty on the relative contribution of ecosystem C fluxes attributable to concurrent meteorological forcing anomalies (concurrent effects) and those attributable to the continuing influence of past phenomena (lagged effects) stifles efforts to explicitly understand the integrated sensitivity of tropical ecosystem to climatic variability. Here we present a conceptual framework—applicable in principle to any meteorology-forced land biosphere model—to explicitly quantify net biospheric exchange (*NBE*) as the sum of anomaly-induced concurrent changes and climatology-induced lagged changes to terrestrial ecosystem C states (*NBE = NBE^{CON} + NBE^{LAG}*). We apply this framework to an observation-constrained analysis of the 2010-2015 tropical C balance: we use a data-model integration approach (CARDAMOM) to merge satellite-retrieved land-surface C observations (leaf area, biomass, solar-induced fluorescence), soil C inventory data and satellite-based atmospheric inversion estimates of $CO_2$ and CO fluxes to produce a data-constrained analysis of the 2010-2015 tropical C cycle. We find that the inter-annual variability of lagged effects explains the majority of NBE inter-annual variability (IAV) throughout 2010-2015 across the tropics (*NBE^{LAG}* IAV = 112% of *NBE* IAV, r = 0.87) relative to concurrent effects (*NBE^{CON}* IAV = 54% of total *NBE* IAV, r = 0.03) and the dominance of *NBE^{LAG}* IAV persists across both wet and dry tropical ecosystems. The magnitude of lagged effect variations on *NBE* across the tropics is largely attributable to lagged effects on net primary productivity (*NPP*; *NPP^{LAG}* IAV 88% of *NBE^{LAG}* IAV, r = -0.99, p-value<0.05), which emerge due to the dependence of *NPP* on inter-annual variations in canopy C mass and





plant-available water states. We conclude that concurrent and lagged effects need to be explicitly and jointly resolved to retrieve an accurate understanding the processes regulating the present-day and future trajectory of the terrestrial land C sink.

## 1 Introduction

Immediate ecosystem responses to external forcings are invariably followed by time-lagged ecosystem responses, attributable

to a continuum of lagged biotic and physical processes. For example, contemporaneous ecosystem state changes attributable to disturbances, climatic variability and increasing atmospheric $CO_2$ levels all induce a temporal spectrum of lagged processes, such as diurnal to seasonal dynamics in canopy and groundwater storage, multi-annual changes in mortality rates, and induce ecosystem dynamics relating to species distributions, nutrient availability and soil properties on timescales spanning from decades to millennia  (Schimel et al. 1997; Smith et al., 2009; Reichstein et al., 2013). Conversely, for a given timespan, the

sum of these "lagged effects" on ecosystem states ultimately represent the ecosystems dynamics attributable to a unique integrated legacy of past phenomena, spanning from diurnal to geologic timescales, making lagged effects a ubiquitous dynamical property of any terrestrial ecosystem. As a consequence, ecosystem function at any given time (such as photosynthetic uptake, respiration and evapotranspiration rates) is an emergent consequence of an ecosystem's initial physical and biotic states and the contemporaneous impact of meteorological forcings on these states.


Disentangling the lagged consequences of past phenomena from contemporaneous impacts of external forcings is a critical priority for understanding and quantifying the contemporary terrestrial carbon (C) cycle responses to climatic variability. Global-scale efforts to resolve the state of the C cycle (Le Quéré et al., 2015) identify tropical C cycle as a dominant contributor to the inter-annual variability (IAV) of the terrestrial C sink. Recent efforts to characterize the tropical C sink IAV have been

largely focused on quantifying the role of concurrent responses to climatic variability, including the contribution of semi-arid ecosystems (Poulter et al., 2014; Ahlstrom et al., 2015), ecosystem responses to drought (Gatti et al., 2014), and more generally continental-scale sensitivities of photosynthesis, respiration and fire fluxes to concurrent temperature and precipitation anomalies (Cox et al., 2013; Andela and van der Werf, 2014; Alden et al., 2016; Jung et al., 2017; Liu et al, 2017; Piao et al., 2019). However, on comparable timescales, time-lagged manifestations of climatic variability on the state of the terrestrial

biosphere have been extensively theorized and observed (Thomson et al., 1996; Schimel et al., 1996, 2005; Richardson et al., 2007; Arnone et al., 2008; Sherry et al., 2008; Saatchi et al., 2013; Frank et al., 2015; Doughty et al., 2015; Baldocchi et al., 2017; Schwalm et al. 2017; amongst many others). Specifically, lagged relationships between climate variability and the terrestrial C fluxes—namely mediated through lagged impacts on photosynthetic uptake and respiration fluxes, groundwater storage, mortality and subsequent shifts of ecosystem function—indicate that lagged effects may be a fundamental component

in the inter-annual evolution of the terrestrial C balance. Observational constraints on terrestrial ecosystem responses to climatic variability further suggest that time-lagged phenomena are a non-negligible component of terrestrial ecosystem C



dynamics on continental-to-global scales (Braswell et al., 1997; Saatchi et al., 2013; Anderegg et al., 2015; Detmers et al., 2015; Fang et al. 2017; Yang et al., 2018). Therefore, while recent efforts to diagnose inter-annual variations of the tropical C balance overwhelmingly emphasize the roles of concurrent forcings, observed ecosystems responses to climatic variability on

multi-annual timescales indicate that the tropical C balance may be substantially affected—if not governed—by lagged responses to inter-annual variations in meteorological and disturbance forcings across tropical ecosystems.

Accurate knowledge of both instantaneous sensitivities and time-lagged processes of terrestrial C cycling to climate is critical for constraining model representations of the terrestrial C cycle. Uncertainty in the long-term terrestrial C flux imbalance and

the associated carbon-climate feedbacks is a prevailing source of uncertainty in Earth System projections (Friedlingstein et al., 2014, Friend et al., 2014), and these are likely underestimated due to a range of under-represented and/or poorly constrained C cycle responses to a changing climate (Luo 2007; Lovenduski & Bonan, 2017). Furthermore, assessments of Earth System projections based on present-day constraints (Cox et al., 2013; Mystakidis et al., 2016) provide little insight on the integrated roles of largely uncertain process controls, including C flux responses to drought (Powell et al., 2013); under-determined C

pool dynamics (Bloom et al., 2016), nutrients dynamics and limitations (Wieder et al., 2015), and higher-order dead organic C dynamics (Schimel et al., 1994, Hopkins et al., 2014). In tropical ecosystems, rapid turnover rates of live and dead organic matter pools, relative to extra-tropical ecosystems (Carvalhais et al., 2014; Bloom et al., 2016) imply interactions between uptake, respiration, and fires (Randerson et al., 2005; Chen et al., 2013, Bloom et al., 2015) on comparable timescales: specifically, given that (a) the mean C residence time in tropical biomass and soil organic matter pools typically spans ~5-50

years, and (b) multi-year observational constraints reveal rapid ecosystem vegetation/C responses to climatic extremes (Saatchi et al., 2013; Alden et al., 2016), sub-decadal timescales are likely a critical for disentangling concurrent and lagged effect impacts on the evolution of tropical C balance. However, despite numerous studies on the roles of productivity (Doughty et al., 2015), water stress (Kurc & Small, 2007; Williams & Albertson, 2004), respiration (Trumbore 2006, Exbrayat et al 2013a,b, Guenet et al., 2018) and mortality (Saatchi et al., 2013, Anderegg et al., 2015; Rowland et al., 2015), there is currently a major

gap between knowledge of individual processes controlling the tropical C balance on inter-annual timescales, and the integrated impact  process interactions leading to complex net C exchanges represented in terrestrial biosphere models (Huntzinger et al., 2013, 2017). As a result, while models provide critical mechanistic insight into complex process interactions, model representations of the net effect of competing and interacting C flux responses to climate variability and disturbance remain highly uncertain on regional and pan-tropical scales. Ultimately, given tropical ecosystems account for 850 Pg of C

and the majority of the Earth's photosynthetic uptake, plant respiration and fire C emissions (Saatchi et al., 2011; Hiederer & Köchy, 2011; Beer et al., 2010; van der Werf et al., 2010), quantitatively understanding the concurrent and long-lived impacts of climatic variability, drought and anthropogenic disturbance is critical for predicting their function in Earth system projections.



Recent inverse estimates of tropical C fluxes from satellite $CO_2$ measurements provide much-needed spatial and temporal constraints on continental-scale Net Biospheric Exchange (NBE; e.g. Takagi et al., 2014, Liu et al., 2014, 2017; Feng et al., 2017., Detmers et al., 2015; amongst others), along with land-surface observations of solar-induced fluorescence (SIF, Frankenberg et al. 2011), leaf-to-soil constraints on total C stocks (Saatchi et al., 2011) and disturbance (Giglio et al., 2013) from land-surface datasets provide a unique opportunity for quantitatively informing terrestrial biosphere model

representations of the tropical C balance. With numerous continental-to-global scale model-data fusion efforts demonstrating the synergistic potential of the present-day "carbon observing system" for resolving the dynamics of the terrestrial C balance (Bloom et al. 2016; MacBean et al., 2018; Exbrayat et al. 2018). Model-data fusion representations of terrestrial ecosystem C cycling ultimately allow for an explicitly mechanistic representation of the terrestrial C balance with in-built states and process parametrizations optimized to represent the observed C cycle variability in the observations; in turn, these terrestrial C balance

models can be used to quantitatively diagnose the concurrent and lagged sensitivities of terrestrial ecosystems to external forcings.

In this study we bring together spatially resolved terrestrial C cycle observations into an ecosystem C balance modelling framework to quantitatively diagnose the role of concurrent and lagged effects on the 2010-2015 inter-annual tropical C

balance. Our analysis is motivated by some key unanswered questions on the large-scale tropical C cycle variability: for instance, are lagged effects significant contributors to inter-annual flux variability on pan-tropical scales? Which C fluxes (e.g. photosynthetic or respiratory) explain the majority of NBE variability attributable to lagged phenomena? Are lagged effects a ubiquitous property across both dry and wet tropical biomes? Here we hypothesize that on a pan-tropical scale, the integrated impact of lagged effects is a critical component of tropical NBE IAV. To test this hypothesis, we reconcile large-scale C cycle

processes and satellite-based estimates of land-to-atmosphere $CO_2$ fluxes using the CARDAMOM diagnostic ecosystem C balance model-data fusion approach. We outline our method in section 2, where we present an analytical methodology for attributing inter-annual ecosystem state variability to concurrent and lagged effects; we present and discuss a quantification of the relative role of concurrent and lagged effects on continental-scale NBE, and the attribution of lagged effects to inter-annual variations in C stock and plant-available water states in section 3; we conclude our manuscript in section 4.


## 2 Methods

To quantitatively diagnose concurrent and lagged effects on the inter-annual tropical C balance, we implement the CARbon DAta-MOdel fraMework (CARDAMOM; Bloom et al., 2016) at a 4°×5° monthly resolution to constrain C cycle fluxes, states

and process controls represented in the Data Assimilation Linked Ecosystem Carbon (DALEC, Williams et al., 2005) model, based on an ensemble of land-surface and atmospheric C cycle observations. The resolution was chosen for consistency with recent estimates of land-surface $CO_2$ and CO fluxes produced at the GEOS-Chem atmospheric chemistry and transport model 4°×5° grid (Bowman et al., 2017; Liu et al. 2017; Jiang et al., 2017). The following section describes the DALEC model (2.1),





satellite and inventory-based observations (2.2), the model-data fusion framework (2.3), and the attribution of the observation-informed DALEC C cycle dynamics to concurrent and lagged effects (2.4). For the sake of brevity, the following sections solely provide a general description of the full model-data fusion implementation (Figure 1); for a complete description of individual models, datasets and methodologies ancillary to our approach, we refer the reader to relevant citations throughout the manuscript. The CARDAMOM analysis is performed across tropical and near-tropical latitudes (30°S - 30°N), and results are regionally evaluated across 6 sub-continental regions, as well as the dry tropics and the wet tropics (Figure A1).

*2.1 Model and drivers*

We use the data-assimilation linked ecosystem carbon model (DALEC; Williams et al., 2005) to represent the principal terms and major pathways of the terrestrial C cycle. The DALEC model family has been extensively used to diagnose terrestrial C cycle dynamics across a range of site level and spatially resolved approaches (Fox et al., 2009; Rowland et al., 2014; Bloom et al., 2016; Smallman et al., 2017; Exbrayat et al., 2018; amongst several others). Here we use DALEC version 2a (henceforth DALEC2a): a summary of the DALEC2a states and processes is depicted in Figure 1. For the sake of brevity, we solely report changes in reference to DALEC2 (previously described by Bloom et al., 2016), and refer the reader to the supplementary material (and references therein) for a complete description of the model.

We extended the DALEC2 structure to include first-order plant-available water pool, where the hydrological balance is defined as the sum of precipitation inputs (P) and evapotranspiration (ET) and runoff (R) outputs. In turn, the plant-available water (W) limits gross primary productivity, through conservation of the inherent water-use efficiency (Beer et al., 2009), where ET is calculated as a function of gross primary production (GPP) and atmospheric vapor pressure deficit (Appendix B1). Effectively, the interaction between W, GPP and ET constitutes a first-order plant-soil carbon-water feedback. We further appended the DALEC2 structure by including a parameterization of soil moisture limitation on heterotrophic respiration (Appendix B2), given that heterotrophic respiration dependence on soil moisture remains highly uncertain (Moyano et al., 2013; Sierra et al., 2015), as well as a dominant source of uncertainty amongst terrestrial C models (Falloon et al., 2011; Exbrayat et al., 2013a,b).

Given a range of in-situ and continental-scale studies highlighting the uncertainties of fire combustion factors across a range of ecosystems, (Ward et al., 1996; Bloom et al., 2015), the errors involved in representing fine-scale fire type variability (Giglio et al., 2013), and spatial variability of fuel loads, we optimize fire C pool combustion factors (in contrast, combustion factors were prescribed as constants in Bloom et al., 2016): specifically, we optimize the combustion factors of foliar biomass ($\pi_{foliar}$), non-foliar biomass pools ($\pi_{nfb}$), soil C ($\pi_{SOM}$) and the fire resilience factor (we approximate the litter C combustion factor as the arithmetic mean of $\pi_{foliar}$ and $\pi_{SOM}$, given that the DALEC2a litter pool represents both above-ground and below-ground C reservoirs). Prior ranges for all $\pi$ and the fire resilience are conservatively defined as spanning 0.01 to 1. We implement the



ecological and dynamic constraints (Bloom & Williams 2015) to ensure that foliar C combustion factors are greater than both non-foliar biomass and soil C combustion factors ($\pi_{foliar} > \pi_{nfb}$ and $\pi_{foliar} > \pi_{SOM}$) which are comprehensively consistent with
detailed measurements of C pool combustion factors across a range of ecosystem fire types (Shea et al., 1996; Araújo et al., 1999, van Leeuwen et al., 2014 amongst others). Finally, we also represent the uncertainty in the longevity of plant labile C; specifically, we now optimize—rather than prescribe—the labile C lifespan used during leaf flushing in DALEC2a (previously all labile C was used during leaf flush, see Bloom & Williams, 2015). The updated model structure is depicted in Figure 1. We henceforth summarize the dynamical description of DALEC2a as


$$\boldsymbol{x}_{t+1} = DALEC2a(\boldsymbol{x}_t, \boldsymbol{M}_t, \boldsymbol{p}) \tag{1},$$

where $\boldsymbol{x}_t$ represents the ecosystem state vector at time $t$, $\mathbf{M}_t$ represents the corresponding meteorological and fire forcings (namely monthly temperature, precipitation, global radiation, vapor pressure deficit, burned area), $\boldsymbol{p}$ represents a vector of
time-invariant process parameters and *DALEC2a()* represents the DALEC2a operation on states $x_t$ throughout time $t \rightarrow t+1$. In summary, DALEC2a optimizable quantities consist of 26 process parameters, $\boldsymbol{p}$, and seven initial ecosystem states (C and $H_2O$ pools; Figure 1) at timestep $t=0$, $\boldsymbol{x_0}$. For the sake of brevity, we include a complete description of DALEC2a state variables, process parameters and diagnostic C fluxes in the supplementary material, except where an explicit mention is necessary in the manuscript.


*2.2 Observations*

The observations assimilated into CARDAMOM are summarized in Table 1. Following Bloom et al., (2016) we assimilate Moderate Imaging Spectroradiometer (MODIS) leaf area index (LAI), soil organic matter (SOM) from the Harmonized world
soil database (HWSD; Hiederer & Köchy, 2011) and above- and below-ground biomass (ABGB, Saatchi et al., 2011). Solar-induced fluorescence (SIF)—retrieved from the Greenhouse Gases Observing Satellite (GOSAT)—is a robust proxy for photosynthetic activity: while non-linear inter-relationships at plant level and flux-tower level have been observed under certain conditions (Verma et al., 2017, Magney et al., 2017), GPP is observed to be linearly inter-related with SIF at ecosystem and regional scales (Frankenberg et al., 2011; Sun et al., 2017). Given that SIF:GPP linear relationships are known to vary
substantially across individual species and entire ecosystems, here we solely assume that monthly SIF provides a constraint on the relative temporal variability of GPP (following MacBean et al., 2018). The averaged 4°×5° SIF values were derived with the polarizations and selection criteria described by Parazoo et al., (2014). The assimilation of relative SIF variability is described in section 2.3.

We assimilate net biospheric C exchange (NBE > 0 for a net biosphere-to-atmosphere flux) estimated using the Carbon Monitoring System Flux atmospheric $CO_2$ inversion framework (CMS-Flux; Liu et al., 2014). In summary, total monthly



$4°\times5°$ surface $CO_2$ fluxes were scaled using a Bayesian 4D variational (4D-Var) inversion approach in order to minimize differences between GOSAT 2010-2013 observations and CMS-Flux representations of total column $CO_2$ (we refer the reader to Liu et al., 2018 for additional details on the derivation of surface $CO_2$ fluxes). Following Liu et al., (2017) and Bowman et

al. (2017), we subtract prior estimates of anthropogenic $CO_2$ emissions from total CMS-Flux total $CO_2$ flux estimates, and we assume that prior anthropogenic $CO_2$ emissions errors are minimal compared to the biospheric $CO_2$ fluxes, given that these are typically much smaller than natural $CO_2$ fluxes at a $4°\times5°$ resolution across the tropics. We use 2015 CMS-Flux NBE estimates constrained by Orbiting Carbon Observatory (OCO-2) total column $CO_2$ observations (Liu et al., 2017) to validate CARDAMOM 2015 NBE estimates and their associated uncertainties (OCO-2 NBE estimates are therefore withheld from the

CARDAMOM NBE assimilation step described in section 2.3).

Finally, we assimilate mean 2010-2015 fire C emission estimates derived from monthly $4°\times5°$ satellite-based estimates of fire CO emissions (Jiang et al., 2017; Worden et al., 2017; Bloom et al., 2019): the estimates of biomass burning CO emissions were derived based on an ensemble of atmospheric CO inversions of column CO measurements from the Measurements of

Pollution in the Troposphere (MOPITT) instrument onboard the NASA EOS/TERRA satellite (Deeter et al., 2014). We refer the reader to Jiang et al., (2017) for the details of the atmospheric CO inversion using the GEOS-Chem adjoint model and to Worden et al., (2017) for the attribution of optimized CO fluxes to biomass burning. Biomass burning CO emission estimates by Worden et al., (2017) were then used to derive total biomass burning C emissions based on monthly estimates of $CO_2$:CO; the approach is detailed in Bowman et al., (2017). We note that NBE estimates exhibit substantial spatial error covariance

structures across individual $4°\times5°$ grid-cells, and the effective information content of the NBE inversions is larger than the $4°\times5°$ resolution. To mitigate the spatial error correlation features identified in the NBE dataset (Bowman et al., 2017; Liu et al., 2017), we employed a $3\times3$ gridcell smoothing window for monthly NBE estimates, following the approach by Liu et al. (2018).

*2.3 Model-data fusion*

Within each $4°\times5°$ grid cell, the C cycle dynamics in DALEC are a function of meteorological drivers $M$, model parameters $p$ and initial conditions $x_0$ (as summarized in Eq. 1). We use a Bayesian inference formulation to independently retrieve the optimal distribution of $x_0$ and $p$ given observations $O$ for each $4°\times5°$ grid cell, where:


$$p(y|O) \propto p(y)p(O|y) \qquad\qquad\qquad (2);$$

$y$ is the control vector $\{x_0,p\}$, p($y$) is the prior probability distribution of $y$, and p($O|y$) is proportional to the likelihood of $y$ given $O$, $L(y|O)$. At any given grid cell, the observation vector $O$ consists of corresponding to LAI, SOM, ABGB, SIF, NBE





and CO-derived fire $CO_2$ emissions (henceforth $O_{LAI}$, $O_{SOM}$, $O_{ABGB}$, $O_{SIF}$, $O_{NBE}$ and $O_{CO}$ respectively), and—assuming errors are uncorrelated— the overall likelihood of $y$ given $O$ can be expressed as

$$L(y|O) = L_{LAI} \, L_{SOM} \, L_{ABGB} \, L_{SIF} \, L_{NBE} \, L_{CO} \qquad (3)$$

For LAI, SOM, ABGB and CO, we derive the corresponding likelihood function $L_*$ (i.e. $L_{LAI}$, $L_{SOM}$, $L_{ABGB}$ and $L_{CO}$, respectively) as follows:

$$L_* = e^{-\frac{1}{2}\Sigma_i\left(\frac{m_i(y)-o_i}{\sigma_i}\right)^2} \qquad (4)$$

where $o_i$ and $m_i(y)$ correspond to the $i$th observation and corresponding modeled quantity derived from control vector $y$, respectively; $\sigma_i$ accounts for the combined effects of DALEC model structural error, model drivers and observation errors. In contrast to Bloom et al., 2016, given that MODIS LAI retrievals have exhibited systematic seasonal biases across the wet tropics (Bi et al., 2015), we solely use mean LAI as a constraint on the mean DALEC2a LAI values (therefore, for the derivation of $L_{LAI}$, $m$ and $o$ in equation 3 correspond to the 2010-2015 mean modeled and observed LAI).


To constrain the relative variability of GPP based on SIF without imposing constraints on the absolute GPP magnitude, we derive $L_{SIF}$—based on Eq.4—by formulating $m$ and $o$ as follows:

$$m_i(y) = \frac{GPP_i}{\overline{GPP}} \qquad (5),$$

$$o_i = \frac{SIF_i}{\overline{SIF}} \qquad (6),$$

where $SIF_i$ and $GPP_i$ are SIF and corresponding DALEC2a GPP values at time index $i$ and $\overline{SIF}$ and $\overline{GPP}$ are the corresponding means during the 2010-2015 time period.

We constrain CARDAMOM NBE using $4°\times5°$ monthly CMS-Flux NBE estimates, derived from GOSAT atmospheric total column $CO_2$ retrievals (Liu et al., 2018) spanning 2010-2013. At each $4°\times5°$ location, we define the $L_{NBE}$ as the product of mean annual NBE and seasonal NBE anomalies using the following equation:

$$L_{NBE} = e^{-\frac{1}{2}\Sigma_a\left(\frac{m_a'(y)-o_a'}{\sigma'}\right)^2} e^{-\frac{1}{2}\Sigma_{i,a}\left(\frac{m_{i,a}''(y)-o_{i,a}''}{\sigma''}\right)^2} \qquad (7),$$






where $m'_a$ denotes annual mean DALEC2a NBE value for year $a$ and $m''_{i,a}$ denote DALEC2a NBE seasonal deviations from their annual means; specifically, for a given month $i$ with corresponding year $a$:

$$m'_a(\boldsymbol{y}) = \frac{1}{12}\sum_{i=1}^{12} NBE_{i,a} \tag{8}$$


$$m''_{i,a}(\boldsymbol{y}) = NBE_{i,a} - m'_a \tag{9}$$

where $NBE_{i,a}$ is the DALEC2a NBE; observations $o'_a$ and $o''_{i,a}$ were derived identically to $m'_a$ and $m''_{i,a}$. The approach outlined in Eq. 7 is employed in order to capture both the seasonal and inter-annual modes of NBE variability: we found that solely minimizing the monthly NBE residuals following the formulation based on Eq. 4 led to disparate inter-annual variations

between the model and observation-constrained NBE. Effectively the formulation in Eq. 7—in comparison to Eq. 4— increases the relative weight of mean annual CMS-Flux NBE constraints on DALEC2a NBE.

The uncertainty for each observational constraint (i.e. $\sigma$ values in Eq. 4 and 7) implicitly represent the combined impacts of observational random errors, systematic errors, and model structural error. In the absence of knowledge on the relative roles

of observation errors in the monthly 4°×5° observation uncertainties and explicit knowledge of model structural error, we prescribed $\sigma$ values through trial and error, in order to (a) ensure that model states and diagnostic variables capture the predominant variability of the observational constraints $\mathbf{O}$, while (b) ensuring that $\sigma$ values are comparable to the observational uncertainty. The uncertainties assumed for each observational constraint are listed in Table 1. For all land surface variables (namely LAI, ABGB, SOM and SIF), $m$ and $o$ were log-transformed (following Bloom et al., 2016). For the mean 2010-2015

LAI constraint, we assumed log-normal uncertainty of $\sigma=\pm\log(1.2)$; we prescribed $\sigma = \pm\log(2)$ log-normal uncertainty structure for each SIF observation. We approximated the uncertainty of the CO-derived mean 2010-2015 fire C values as $\sigma = \pm\ 20\%$, which is broadly consistent with the monthly 4°×5° CO uncertainty estimates and the corresponding $CO_2$:CO uncertainty estimates reported by Bowman et al., (2017) and Worden et al., (2017). For NBE we prescribed $\sigma' = 0.02$ gC/m2/day and $\sigma''$ = 2 gC/m2/day; we found that these were suitable to capture the first-order 2010-2013 seasonal and inter-annual components

of continental scale NBE variability. We discuss the potential impacts of observation uncertainty assumptions and make recommendation for future efforts in section 3.4.

To retrieve the distribution of p($\boldsymbol{y}$|$\boldsymbol{O}$), we employed an adaptive metropolis-Hastings Markov Chain Monte Carlo (MHMCMC) approach following Bloom et al., (2016). We generally found that the computational costs required to meet MHMCMC

convergence criterion reported by Bloom & Williams (2015) for each 4°×5° grid-cell were prohibitively expensive. We updated the adaptive MHMCMC to the Haario et al., (2001) MHMCMC approach, where the MHMCMC proposal distribution is adapted as a function of previously accepted samples (see Haario et al., 2001 for algorithm details). We ran 4 adaptive MHMCMC chains for $10^8$ iterations in each 4°×5° grid-cell. We found that the latter half of the chains converged within a



Gelman-Rubin convergence criterion value of <1.2 in 75% of the grid cells. For the subsequent analysis, we a subset of 500

samples of $y$ from the of latter half of each MHMCMC chain, totaling 4×500 samples of $y$ per 4°×5° grid-cell.

*2.4 Concurrent and lagged effects*

Ecosystem function—such as photosynthesis, respiration and evapotranspiration rates—at all stages of ecological succession

is both a consequence of an ecosystem's initial physical and biotic states and the contemporaneous impact of meteorological forcings on these states. For example, ecosystem water and nutrient availability along with species demography and species composition—effectively amounting to the time-integrated ecosystem legacy—will govern an ecosystem's function under a nominal forcing. The cumulative impact of both episodic or prolonged variability in external forcings will be "remembered" in ecosystem states, thus shaping ecosystem function as an emergent property of external forcing history. Ecosystem states

under a constant and perpetual environmental forcing will follow a trajectory towards an equilibrium state (as has been largely hypothesized as the typical outcome for ecosystem C stocks; Luo and Weng 2011, Luo et al. 2015) or more generally a transient trajectory about a domain of attraction (Holling, 1973), with stable equilibria, stable limit cycles, stable nodes and/or neutrally stable orbits as potential trajectories. Here, we define *lagged effects* as the sum of ecosystem state changes induced by a reference climatological mean forcing (Figure 2); these include the functional responses of ecosystem under climatological

conditions (e.g. joint photosynthesis, respiration and evapotranspiration responses to non-equilibrium plant-available water, leaf area, biomass and dead organic C states), as well as functional shifts (e.g. succession-induced changes in demography and species composition, and consequently changes in ecosystem-scale photosynthetic capacity). In addition to an attraction towards a fixed equilibrium or domain, ecosystem states are perpetually disturbed by exogenous forces, such as meteorological forcing anomalies relative to a climatological mean forcing. Here we define these *concurrent effects* as all anomaly-concurrent

changes to ecosystem states unaccounted for by climatology-induced state changes (i.e. *lagged effects*); these include functional responses to anomalous forcings (e.g. drought impact on photosynthetic uptake and respiration in responses to meteorological phenomena), as well as functional shifts on demographics and species composition induced by concurrent mortality and disturbance events.  On an annual basis, the total state changes resulting from both concurrent and lagged effects will in turn propagate into future ecosystem states. In this manner, forcing anomalies are perpetually propagated into ecosystem

states, and lagged effects in subsequent years represent an integrated legacy of all prior phenomena. The choices of (a) "concurrent effects" to describe effects contemporaneous to a meteorological event and (b) "lagged effects" to describe all time-lagged processes are consistent with Frank et al., (2015) definitions associated with effects occurring during or after a climatic anomaly. While in this study we confine our analysis to concurrent and lagged effects on inter-annual timescales, we note that this conceptual framework can in be principle be adapted to diagnose physical and biological ecosystem state changes

on any timescale of relevance.





Here we present a dynamical formulation for the derivation of concurrent and lagged effects on the inter-annual ecosystem state changes. To explicitly quantify the concurrent effects and lagged effects, we define the trajectory of the modeled dynamic state variables $x$ at year $a+1$ as


$$x_{a+1} = D(x_a, M_a, p) \tag{10},$$

where the state vector $x_{a+1}$—which is comprised of DALEC2a states at the beginning of year $a+1$—is computed from the DALEC2a model operator $D()$, which is a function of the previous state $x_a$ at beginning of year $a$, the meteorological forcing

history of the previous year $M_a$, and time-invariant ecosystem parameters $p$. We note that Eq. 10 is resolved on an annual time-step; however, the DALEC2a operator time-step is monthly, hence the operator in Eq. 10 is a composite of monthly operators as denoted in equation 1. To isolate the role of concurrent meteorological and disturbance anomalies in $M_a$, we define the C trajectory under a reference climatological mean forcing $M'$ as

$$x'_{a+1} = D(x_a, M', p) \tag{11}.$$

Here we define $M'$ as the monthly climatological mean of the 2010-2015 meteorological drivers, where

$$M_a = M' + \delta M_a. \tag{12}.$$


With Eq. 10 and 11, we can define the change in the state $x$ in year $a$, $\delta x_a$, as

$$\delta x_a = x_{a+1} - x_a = (x_{a+1} - x'_{a+1}) + (x'_{a+1} - x_a) \tag{13}.$$

This formulation allows us to define the lagged effect on ecosystem states in year $a$ as

$$\delta x_a^{LAG} = x'_{a+1} - x_a$$

$$\tag{14},$$

and the concurrent effect on ecosystem states in year $a$ as


$$\delta x_a^{CON} = x_{a+1} - x'_{a+1} \tag{15};$$

and the sum of concurrent and lagged effects in Eq. 14 and 15 as





$\delta \boldsymbol{x}_a = \delta \boldsymbol{x}_a^{LAG} + \delta \boldsymbol{x}_a^{CON}$                                                                          (16).

We conceptually illustrate the derivation of annual concurrent and lagged effects on a given ecosystem state $x$ in Figure 3. Under a climatological mean forcing (blue line), the ecosystem state trajectory—solely induced by lagged processes—would diverge from meteorology-forced ecosystem state trajectory (black line), and would eventually converge to an equilibrium state or oscillate about a domain of attraction (Figure 3a). For a one-year timespan, the change in ecosystem state $x$ throughout year $a$, $\delta \boldsymbol{x}_a$ can be decomposed into a climatology-induced lagged effect change $\delta \boldsymbol{x}_a^{LAG}$, and an anomaly-induced concurrent effect change $\delta \boldsymbol{x}_a^{CON}$ (Figure 3a, inset).

From a mechanistic standpoint, the variability of $\delta \boldsymbol{x}_a^{LAG}$ is independent of meteorological forcing anomalies and is therefore solely dependent on all ecosystem states $x_a$. For example, in a hypothetical scenario where a climatological mean forcing induces no net ecosystem state changes, then $\delta \boldsymbol{x}_a^{LAG} = \boldsymbol{x}_a - \boldsymbol{x'}_{a+1} = \boldsymbol{0}$, and $\delta x = \delta x^{CON}$. In a more general scenario, $\delta \boldsymbol{x}_a^{LAG} = \boldsymbol{x}_a - \boldsymbol{x'}_{a+1} \sim constant$ for all $a$: in this instance $x_a^{LAG}$ is non-zero but largely insensitive to variations in $x_a$ within a typical range of ecosystem states $\boldsymbol{x}$, therefore (i) the year-to-year variability of $\delta \boldsymbol{x}$, is largely dependent on the variability of $\delta \boldsymbol{x}^{CON}$, and (ii) $\delta \boldsymbol{x}^{LAG}$ amounts to an approximately constant offset term (Figure 3b). Alternatively, if $\delta \boldsymbol{x}^{LAG}$ is sufficiently sensitive to the variability of $\boldsymbol{x}$, the variability of $\delta \boldsymbol{x}$ will be a function of both $\delta \boldsymbol{x}^{LAG}$ and $\delta \boldsymbol{x}^{CON}$: in this instance, year-to-year variations in $\boldsymbol{x}$ are influencing both the sign and magnitude of lagged effects (Figure 3c).

Here we investigate the possible contributions of the annual variability of $\delta \boldsymbol{x}^{CON}$ and $\delta \boldsymbol{x}^{LAG}$ on $\delta \boldsymbol{x}$ for the 2010-2015 time period across tropical ecosystems. Specifically, we test the two following hypotheses:


- _Hypothesis 1_: $var(\delta \boldsymbol{x}^{LAG}) \ll var(\delta \boldsymbol{x}^{CON})$. In this instance, the impact of $\boldsymbol{M'}$ on $\boldsymbol{x}$ is largely independent on the variability of $\boldsymbol{x}$; consequently the year-to-year variability of the lagged effects force $\delta \boldsymbol{x}^{LAG}$ is relatively small, and the year-to-year changes in ecosystem states, $\delta \boldsymbol{x}$, are dominated by $\delta \boldsymbol{x}^{CON}$ (Figure 3b).

- _Hypothesis 2_: $var(\delta \boldsymbol{x}^{LAG}) \sim var(\delta \boldsymbol{x}^{CON})$. In this instance, the impact of $\boldsymbol{M'}$ on $\boldsymbol{x}$ is dependent on the variability of $\boldsymbol{x}$; consequently, the year-to-year variability of the lagged effects $\delta \boldsymbol{x}^{LAG}$ is substantial, and the year-to-year changes in ecosystem states, $\delta \boldsymbol{x}$, are substantially attributable to both $\delta \boldsymbol{x}^{CON}$ and $\delta \boldsymbol{x}^{LAG}$ (Figure 3c).

The mechanistic nature of the DALEC2a model within CARDAMOM (namely the representation of allocation fractions, residence times, meteorological sensitivities and explicit representation of dynamical states) allows for a data-constrained probabilistic assessment of the relative role lagged and concurrent effects on net ecosystem state changes. In principle, the disaggregation of $\delta \boldsymbol{x}_a$ into $\delta \boldsymbol{x}_a^{CON}$ and $\delta \boldsymbol{x}_a^{LAG}$ (and the associated hypotheses 1 and 2) can in principle be projected onto any subset of net ecosystem fluxes or additive combination of gross fluxes. For example, the NBE in year $a$ ($NBE_a$) corresponds





to the net C loss between $x_a$ and $x_{a+1}$; in turn, $NBE_a$ can be decomposed into its lagged effect component ($NBE_a^{LAG}$) and the concurrent effect component ($NBE_a^{DIR}$), where


$$NBE_a = NBE_a^{CON} + NBE_a^{LAG} \tag{17};$$

$NBE_a$ and $NBE_a^{LAG}$ can be directly calculated from $D(x_a, M_a, p)$ and $D(x_a, M', p)$ respectively, and $NBE_a^{CON}$ is calculated as $NBE_a - NBE_a^{LAG}$. By definition in the DALEC2a model, NBE is the sum of primary productivity (NPP), heterotrophic

respiration (RHE) and fire (FIR) fluxes, where:

$$NBE_a = RHE_a + FIR_a - NPP_a \tag{18}.$$

In turn, disaggregation *RHE*, *FIR* and *NPP* into their respective concurrent and lagged components gives:


$$NBE_a^{CON} = RHE_a^{CON} + FIR_a^{CON} - NPP_a^{CON} \tag{19}.$$
$$NBE_a^{LAG} = RHE_a^{LAG} + FIR_a^{LAG} - NPP_a^{LAG} \tag{20}.$$

To diagnose relative inter-annual variations of a given flux $F$ (namely the 2010-2015 timeseries of *NBE*, *RHE*, *FIR* and *NPP*),

we derive annual anomalies $\Delta F$ relative to the mean 2010-2015 flux $\overline{F}$, where for a given year $a$:

$$\Delta F_a = F_a - \overline{F} \tag{21}.$$

The $\Delta$ operation in Eq. 21 can be implemented onto each term in Eq. 18-20 without loss of equivalence between left-hand and

right-hand sides (for example, $\Delta NBE_a^{LAG} = \Delta RHE_a^{LAG} + \Delta FIR_a^{LAG} - \Delta NPP_a^{LAG}$).

Finally, we diagnose the 2010-2015 $\Delta NBE_a^{LAG}$ variability as a function of the inter-annual anomalies in individual ecosystem states, $\Delta x_{a(*)} = \{ \Delta x_{a(1)}, \ \Delta x_{a(2)}, \dots, \Delta x_{a(N)} \}$, relative to the mean ecosystem state $\overline{x}$. For DALEC2, these consist of annual anomalies in initial C and $H_2O$ states (see Figure 1). For a given year, the total NBE lagged effect anomaly, $\Delta NBE_a^{LAG}$ can be

decomposed into

$$\Delta NBE_a^{LAG} = \sum_{n=1}^{N} \Delta NBE_{a(n)}^{LAG} + \Delta I_a \tag{22};$$

$\Delta NBE_{a(n)}^{LAG}$ represents the NBE lagged effect component solely attributable to an anomaly in ecosystem state $n$ ($\Delta x_{a(n)}$), and

$\Delta I_a$ collectively accounts for the contribution of higher-order interactions between individual ecosystem states. In other words,





given that $\Delta NBE_a^{LAG}$ is solely attributable to variability of annual initial conditions $x_a$, the decomposition of $\Delta NBE_a^{LAG}$ to individual pool contributions provides a first-order attribution of lagged effect IAV to underlying C and H$_2$O pool dynamics. The derivation of Eq. 22 is explicitly described in Appendix C.

To derive uncertainty estimates for each annual flux $F_a$, or corresponding anomaly $\Delta F_a$, we calculate each term based on the 2000 samples of $y$ at each gridcell (see section 2.3), and we calculate the corresponding median and inter-quartile range (25$^{th}$-75$^{th}$ percentiles) for each term. Inter-annual variations in 2010-2015 $F$ and $\Delta F$ timeseries are reported as standard deviations. We conservatively assume that these errors are fully correlated when propagating these uncertainties across each region.

## 3. Results and Discussion

### 3.1. Evaluation of observation-constrained tropical C balance

Ultimately inferences about the concurrent and lagged effects on NBE can only be drawn if CARDAMOM is able to both (i)
accurately represent observed NBE, and (ii) accurately predict the underlying processes controlling IAV. Here we evaluate the ability of CARDAMOM to (a) capture the assimilated NBE dataset (2010-2013), and (b) predict the observed OCO-2 derived 2015 NBE. Optimized CARDAMOM NBE (a function of the optimized DALEC2a parameters and initial 2010 ecosystem states) broadly represents the monthly variability of the 2010-2013 regional-scale assimilated GOSAT-retrieved NBE (Figure 4; Table 2). In individual regions, monthly CARDAMOM versus CMS-Flux NBE r $\geq$ 0.75, with the exception of South-East
Asia and Indonesia region (r = 0.57) where the CARDAMOM and GOSAT-retrieved NBE exhibits a relatively small seasonality compared to other regions. Evaluation of CARDAMOM NBE against withheld NBE estimates from OCO-2 exhibit a degradation in the correlation and RMSE values, but agree favorably on the amplitude and timing of the NBE variability (Table 2). We find that the CARDAMOM analysis is able to robustly capture the 2010-2013 GOSAT-derived NBE IAV at regional scales (see Figure 5 & Table 2; regional NBE r $\geq$ 0.95). On an annual basis, the 3 out of 6 regional OCO-2 annual
NBE estimates 2015 are within 90% CARDAMOM prediction confidence intervals; all OCO-2 annual NBE estimates are within CARDAMOM 2015 prediction confidence intervals for the wet tropics, dry tropics and the entire tropical study region (Figure 5). We also note that regions the confidence intervals of the 2014-2015 predictions are substantially larger than uncertainty intervals within the 2010-2013 analysis period, likely due to under-constrained modes of long-term terrestrial C cycle variability.


The spatial variability of CARDAMOM state variables and fluxes constrained by static datasets, namely LAI, biomass, soil C and mean fire C emissions (Table 1), are broadly correlated with the observational constraints by the CARDAMOM analysis (r = 0.67 – 0.99; p<0.05; Figure S1); for the above-mentioned quantities total biases amounted to <10%. The correlation between CARDAMOM GPP and GOSAT SIF is positive & significant (p-value <0.05) in 74% of 4x5 pixels, with higher





465 correlations in the dry tropics (25$^{th}$ – 75$^{th}$ percentile = 0.41 – 0.78) relative to the wet tropics (25$^{th}$ – 75$^{th}$ percentile = 0.16 – 0.66); the lower correlations in the wet tropics are to be expected, given that wet tropical ecosystems fundamentally exhibit a weaker GPP seasonal cycle. In general, we argue that (i) CARDAMOM terrestrial C cycle dynamics broadly represent the variability of assimilated C cycle datasets, and (ii) CARDAMOM NBE and associated uncertainties compare favorably against withheld data on seasonal and inter-annual timescales (albeit with limited annual NBE skill at regional scales). We anticipate

470 that the ever-growing satellite $CO_2$ record—along with increasing volume and quality of terrestrial C cycle observations—will ultimately lead to improved seasonal and inter-annual process representations in future model-data fusion analyses of the terrestrial C balance.

*3.2 Concurrent and lagged effects on the tropical C balance*


The attribution of annual $\Delta NBE$ into its concurrent and lagged components ($\Delta NBE^{CON}$ and $\Delta NBE^{LAG}$) reveal that both are substantial contributors to regional and pan-tropical $\Delta NBE$ (Figure 6). On a regional scale, $\Delta NBE^{CON}$ IAV and $\Delta NBE^{LAG}$ IAV during 2010-15 amount to 35-108% and 43-159%, relative to $\Delta NBE$ IAV (Table 3). Notable $\Delta NBE^{CON}$ anomalies include the positive $\Delta NBE^{CON}$ values in both South America regions during the 2010 South America drought, and negative $\Delta NBE^{CON}$

480 values during the relatively wet 2010-2011 conditions in Australia, which corroborates the generally hypothesized responses of tropical ecosystems to wet and dry extreme events (Lewis et al., 2011; Bastos et al., 2013). In the large majority of cases— both on a regional and pan-tropical scale—the sign of the year-to-year $\Delta NBE$ changes and the $\Delta NBE^{CON}$ changes are consistent: for example, the sign of 2010-to-2011 $\Delta NBE$ and $\Delta NBE^{CON}$ changes are consistent in all 6 regions, and in aggregate across the whole tropics (Figure 6). In contrast, while magnitude of the year-to-year $\Delta NBE^{LAG}$ changes is comparable to $\Delta NBE^{CON}$, the

485 year-to-year $\Delta NBE^{LAG}$ and $\Delta NBE$ changes are generally uncorrelated. In contrast, the prominent role of $\Delta NBE^{LAG}$ on a regional and pan-tropical $\Delta NBE$ is manifested through the long-term trends in $\Delta NBE^{LAG}$, resulting in a positive correlation across all regions between 2010-2015 $\Delta NBE$ and $\Delta NBE_{LAG}$. $\Delta NBE^{LAG}$ IAV exceeded $\Delta NBE^{CON}$ IAV across all regions except South-East Asia and Indonesian archipelago (Table 3). $\Delta NBE^{LAG}$ IAV values amounting to >100% of $\Delta NBE$ IAV are attributable to regional and pan-tropical anti-correlations between $\Delta NBE^{LAG}$ and $\Delta NBE^{CON}$: specifically, $\Delta NBE^{LAG}$ and $\Delta NBE^{CON}$ are

490 anticorrelated across the tropics (r = -0.46), and all regions except Australia and SE Asia & Indonesia (r = -0.75–0.58); while none of the correlations are significant within a p-value<0.05, the consistent anticorrelation across all regions suggests that lagged effects may significantly and systematically dampen the impact of $\Delta NBE^{CON}$ on longer timescales. On a pan-tropical scale, we found that the $\Delta NBE^{LAG}$ IAV exceeded $\Delta NBE^{CON}$ IAV, and the relative prominence of $\Delta NBE^{LAG}$ IAV persisted across both the dry tropics and the wet tropics (Table 3, regions denoted in Figure A1), suggesting that $\Delta NBE^{LAG}$ is a ubiquitous

495 component of inter-annual tropical C cycle dynamics.

The decomposition of $\Delta NBE^{CON}$ into constituent fluxes ($\Delta NPP^{CON}$, $\Delta RHE^{CON}$ and $\Delta FIR^{CON}$) reveals that a diverse set of processes drive the concurrent NBE response to meteorological forcing (Figure 7, middle column). For four out of six regions,



$\Delta NPP^{CON}$ variability accounts for the largest contribution to $\Delta NBE^{CON}$ variability (Table 4), however $\Delta RHE^{CON}$ and $\Delta FIR^{CON}$

contributions area magnitudes are comparably substantial. In northern hemisphere South America, $\Delta RHE^{CON}$ variability

accounts for the largest contribution to $\Delta NPP^{CON}$ suggesting that to first order the integrated impacts of meteorological forcing

IAV on respiration outweigh the year-to-year impacts on photosynthetic uptake. In Australia, the concurrent impact of fires

on $\Delta NBE^{CON}$ is largest but comparable to $\Delta NPP^{CON}$ (Table 4). In contrast, the decomposition of $\Delta NBE^{LAG}$ into constituent

fluxes ($\Delta NPP^{LAG}$, $\Delta RHE^{LAG}$ and $\Delta FIR^{LAG}$), reveals that $\Delta NBE^{LAG}$ is ubiquitously dominated by $\Delta NPP^{LAG}$ variability (55-127%),

followed by modest contributions from $\Delta RHE^{LAG}$ variability (12-46%) and minimal contributions by $\Delta FIR^{LAG}$ variability (0-

3%, see Table 4). The prominence of $\Delta NPP^{LAG}$ is attributable to faster continental-scale response of C uptake following year-

to-year variations in initial C and $H_2O$ states (relative to $\Delta RHE^{LAG}$), indicating that biomass pool dynamics (rather than dead

organic C states) dominate initial ecosystem responses to external forcing anomalies. The relatively small contribution of

$\Delta FIR^{LAG}$ values to $\Delta NBE^{LAG}$ indicate that the magnitude of fires is, to first order, dominated by variability in the forcing, rather

than variability in the states within fire-prone ecosystems.

We find that variability in foliar C, plant-available $H_2O$ and soil C contribute to the majority of regional and pan-tropical

$\Delta NBE^{LAG}$ variability (Figure 8). For example, the enhanced foliar C in 2011 over the Australian continent (relative to 2010)—

attributable to a combination of reduced fires and increase productivity due to anomalous 2010 conditions over the Australian

continent (Figure S2)—alone contributed to a 0.1PgC/yr net uptake increase (i.e. NBE reduction) relative to 2010. We find

that the sum of all the pool-specific $\Delta NBE^{LAG}$ anomalies approximately add up to $\Delta NBE^{LAG}$ (Figure S3), indicating that—

insofar as these are represented in DALEC2a—$\Delta NBE^{LAG}$ is (a) to first order equivalent to the sum of $NBE^{LAG}$ sensitivities to

individual initial states, and (b) cross-pool interactions ("$I$" in Eq. 22) are a secondary component of $\Delta NBE^{LAG}$. In aggregate,

we find that foliar C variability contributes to 31-82% of $\Delta NBE^{LAG}$ variability across all regions, and 47% of the pan-tropical

$\Delta NBE^{LAG}$. Southern Africa and Australia are the only regions where inter-annual variations in soil C and plant-available water

(respectively) contribute to more variability than foliar C (Table 5). In other words, our results indicate that under a

climatological mean forcing, (a) year-to-year changes in foliar C and soil water initial conditions are sufficient to induce

substantial year-to-year changes in C uptake, and (b) year-to-year changes in soil C are sufficient to substantially influence

total heterotrophic respiration rates. We find that the remaining states (labile C, wood C, fine root C and litter C) explain < 0.2

PgC/yr variability of $\Delta NBE^{LAG}$ across all regions. The gradual increase of $\Delta NBE^{LAG}$ across all tropical regions (Figure 6) is

jointly attributable to changes in soil C and foliar C, while plant-available water exhibits no substantial trend: these results

suggest that tracking the long-term evolution of tropical ecosystem canopy cover (Saatchi et al., 2013; Shi et al., 2017) and

reducing the process-level uncertainties associated with foliar C dynamics relationships to meteorological forcings (discussed

in 3.4) are potentially critical for advancing quantitative understanding of tropical NBE IAV. We also note that, while our

analysis focused on the $\Delta NBE^{LAG}$ sensitivity to year-to-year ecosystem states changes, the magnitude of $\Delta NBE^{CON}$ is also in

principle dependent on time-varying ecosystem states (Figure 2). We therefore highlight that further efforts to quantitatively

establish the sensitivity of $\Delta NBE^{CON}$ on year-to-year ecosystem state changes would amount to a critical step towards (a) better



resolving the present and evolving function of the terrestrial C balance, and (b) quantitatively characterizing the cumulative impact of climate anomalies (e.g. larger and/or more frequent droughts) on the net accumulation of C in terrestrial ecosystems.


The predominant influence of $\Delta NPP^{LAG}$ on $\Delta NBE^{LAG}$ is manifested itself as both (i) inter-annual variations in $\Delta NPP^{LAG}$, such as the 2011 response to 2010 wet conditions in Australia and dry conditions in South America, and (ii) secular declines in $\Delta NPP^{LAG}$ across most tropical regions (Figure 7). The above-mentioned $\Delta NPP^{LAG}$ anomalies in 2011, relative to 2010 $\Delta NPP^{LAG}$, amount to lagged responses attributable to the 2010 meteorological impacts on ecosystem states. In contrast, the

secular trends in $\Delta NPP^{LAG}$, as well as the contributions of foliar C and plant-available water to $\Delta NBE^{LAG}$, suggest a progressive shift in ecosystem states may be gradually altering the magnitude of $\Delta NBE^{LAG}$ and consequently $\Delta NBE$. The difference in $\Delta NPP^{LAG}$ and $\Delta NBE^{LAG}$ between 2010 and 2015 (in both cases > 2PgC/yr) suggests either a prolonged response to 2010 forcings and/or a longer-term decline in vegetation's photosynthetic capacity. Together the above mechanisms suggest that lagged impacts of changes photosynthetic capacity (due to either changes in canopy or water availability) operate at multiple

timescales but are nonetheless a fundamental component of tropical NBE IAV. In contrast, lagged heterotrophic respiration responses ($\Delta RHE^{LAG}$) are a secondary component of $\Delta NBE^{LAG}$, due to the inherent lags of biomass growth and subsequent mortality, combined with ~5-50yrs mean dead organic C residence times across tropical ecosystems (Bloom et al., 2016). Field-scale measurements and continental-scale inferences of NPP responses to past climatic anomalies highlight the potential importance of lagged NPP effects (Sherry et al., 2008, Detmers et al., 2015; Wolf et al., 2016), and prolonged impact of the

2005 drought on canopy water content (Saatchi et al. 2013) suggests that post-disturbance NPP decline could persist for multiple years. Continued monitoring of NBE (e.g. following the 2015-2016 ENSO event), and subsequent attribution to concurrent and lagged effects, will be critical to better quantify the multi-year NPP recovery (e.g. Schwalm et al., 2017) and to improve confidence in secular NPP changes on the tropical C balance attributable to long-term shifts in ecosystem C states.

*3.4 Observation and model uncertainties and caveats*

The prescribed observation uncertainty characteristics (Table 1) are potentially a critical source of error in the data-informed representation of terrestrial C cycle dynamics and its subsequent partitioning into concurrent and lagged effects. For example, relative differences in the mean NBE values retrieved from aircraft and satellite $CO_2$ measurements over the Amazon Basin

(Alden et al., 2016; Bowman et al. 2017) highlight the need to determine the sensitivity of our results to top-down estimates of NBE. While the uncertainty structures of top-down $CO_2$ inversion estimates is beyond the scope of our paper, we recognize the need to robustly assess and characterize uncertainties in seasonal and inter-annual variations in NBE. Potential limitations in the linear SIF:GPP assumption include (i) systematic underestimations of afternoon GPP stress, given that the GOSAT overpass time is ~1pm, and (ii) uncharacterized biases emerging from non-linear SIF:GPP under extreme conditions (Verma

et al., 2017). We highlight that recent efforts to merge multiple SIF datasets (Zhang et al., 2018), and process-based representations of SIF:GPP (Bacour et al., 2019) can together be used to improve the accuracy of SIF:GPP representation in



CARDAMOM. We generally acknowledge that more elaborate approaches and a more comprehensive treatment of model and data error characteristics are necessary to understand the contribution of individual data streams error (Keenan et al., 2011; Heald et al., 2004; MacBean et al., 2016, 2018). Specifically, the explicit and accurate representation of model structural error

is critical for both accurate retrievals of physical parameters and accurate model predictions (Brynjarsdottir & O'Hagan, 2014) and solving for error model parameters (Schoups & Vrugt 2010, Xu et al., 2017) is potentially advantageous for physical parameter retrievals and prediction purposes. For example, we note that without an error model structure we cannot explicitly account for cross correlations in the errors between observations or the impacts of heteroscedasticity (Schoups & Vrugt, 2010). While the identification and optimization of an appropriate structural error model is beyond the scope of this manuscript, we

highlight this as an important priority for future CARDAMOM analyses.

Unrepresented processes DALEC2a model structure—particularly processes that are potentially substantial contributors to $\Delta NBE^{CON}$ and $\Delta NBE^{LAG}$—amount to an additional source of uncertainty in our analysis. Potentially critical processes include time-varying autotrophic respiration (Rowland et al., 2014), plant C allocation and plant mortality, as well as explicit

representation of coarse woody debris (Smallman et al., 2017). In particular, given that our results suggest that foliar C is a major contributor to $\Delta NBE$, unrepresented processes relating to tropical leaf phenology may substantially impact the accuracy of lagged effect attribution, including phenological processes regulating leaf onset, leaf lifespan and litterfall seasonality (Chave et al., 2010; Caldararu et al., 2013, Xu et al., 2016), as well as the time-varying allocation regimes (Doughty et al., 2015). Furthermore, while the DALEC2a phenology assumes a time-invariant ratio between LAI and foliar C (i.e. a time-

invariant ecosystem-level leaf carbon mass per area), the joint roles of leaf demographics and species distribution on the temporal variability of leaf carbon mass per area could potentially amount to a significant impact on photosynthetic capacity, and subsequently on the variability of $\Delta NBE^{CON}$ and $\Delta NBE^{LAG}$. We also highlight year-to-year changes in species composition (such as $C_3:C_4$ plants) and the temporal dynamics of vegetation and soil nutrients as potential contributors to $\Delta NBE^{LAG}$ (Sherry et al., 2008; Schimel et al., 1997) as potentially unrepresented but critical processes, particularly in fire-prone regions

(Pellegrini et al., 2018) and nutrient-limited tropical forest ecosystems (Wieder et al. 2015). Finally, we highlight the need to investigate the sensitivity of our results to the 2010-2015 climatological mean forcing: while to first order the diagnosis of lagged effect anomalies from the mean (rather than absolute values) are insensitive to the reference forcing, further efforts are required to determine whether non-linear impacts of an alternative reference forcing (e.g. a climatological mean forcing based on a 30-year climate normal) may amplify or dampen $\Delta NBE^{LAG}$ IAV estimates.


Our continental-scale results indicate that DALEC2a model complexity is adequate to both represent NBE variability and accurately predict NBE outside the training window on a pan-tropical scale (2015), which provides a first-order assessment of the adequacy of the DALEC2a model structure. However, we note considerable biases in 2015 NBE estimates in individual regions (including Australia and Northern Hemisphere South America), which may suggest that either (a) the model structure

is significantly biased or (b) the observational constraints are insufficient to make accurate predictions. To determine the relative



impact of model error, we anticipate that additional insights could be obtained by retrieving $\Delta NBE^{CON}$ and $\Delta NBE^{LAG}$ based on alternative DALEC model structures (Fox et al., 2009; Smallman et al., 2017). For example, the implementation of DALEC2a assimilation and prediction evaluation across long-term records eddy covariance $CO_2$ and $H_2O$ fluxes would amount to a useful evaluation of the model structure constrained by multiple data streams (e.g. following Richardson et al., 2010; Keenan et al.,

2013; Smallman et al., 2017), though few tropical ecosystem sites where multi-year NBE constraints are available. Furthermore, to diagnose the potential role of higher-order process interactions on lagged and concurrent effects—such as nutrient limitations, ecosystem demography and explicit representations of carbon-water-energy interactions—we highlight that the $\Delta NBE^{CON}$ and $\Delta NBE^{LAG}$ attribution methodology introduced here can in principle be applied using higher complexity terrestrial biosphere models (e.g. Huntzinger et al., 2013, 2017; Macbean et al., 2018; Longo et al., 2019).


**4. Conclusions**

The prominent role of $\Delta NBE^{LAG}$ across the tropics throughout 2010-2015 supports our second hypothesis (section 2.4), namely that concurrent and lagged effect variations are comparable on inter-annual timescales. By constraining a diagnostic ecosystem

C balance model with an array of terrestrial C cycle observations (LAI, biomass, soil C, SIF, CO-derived fire C emissions and CO2-derived NBE), we show that even though 2010-2015 $\Delta NBE^{CON}$ account for a considerable variability of NBE during exceptionally dry and wet conditions, $\Delta NBE^{LAG}$ accounts for the majority of the 2010-2015 variability across the tropics. While 2010-2015 $\Delta NBE^{CON}$ is accounted for by a relatively even mix of fires, heterotrophic respiration and NPP responses to climatic variability, $\Delta NBE^{LAG}$ variability is overwhelmingly dominated by the impact of inter-annual variations in lagged NPP effects,

followed by a modest contribution from the state-dependence of heterotrophic respiration. In aggregate, anomalies in plant-available water, foliar C and soil C were identified as the primary influences on $\Delta NBE^{LAG}$ variability. Our findings therefore highlight a critical need to explicitly account for lagged effects when investigating the process-level tropical NBE responses to climatic variability on inter-annual timescales. Furthermore, we highlight the need to accurately and continuously resolve NBE at sub-continental scales in order to advance our mechanistic and process-level understanding of terrestrial C cycling and

its evolving sensitivity to climate.

**Appendix A: Regional definitions**

**Appendix B: Model description**

The following sections provide a summary of the process parameterizations introduced in the DALEC version implemented in the Bloom et al., (2016) study. For completeness, a full description of DALEC2a is provided in the manuscript supplement.





**B1. DALEC2a Water balance and GPP water stress**

The DALEC2a plant-available water balance at timestep $t+1$ in is derived as

$$W_{t+1} = W_t + (P_t - R_t - ET_t)\,\Delta t \qquad\text{(B1),}$$


where W denotes total plant-available water [in mm water storage equivalent], and $P$, $R$ and $ET$ precipitation, runoff and evapotranspiration fluxes [mm/day] over the time period $\Delta t$ [days]. We note that this equation represents a water balance in the dynamic, plant-available water pool and does not include deep groundwater, confined aquifers or other unconnected/static storages. Following a generalized non-linear reservoir formulation, we parameterize monthly runoff losses as a second-order

decay function with respect to storage, $W_t$, as:

$$R_t = \alpha W_t^2 \qquad\text{(B2),}$$

where $\alpha$ is a second-order decay constant [mm$^{-1}$ day$^{-1}$]. The dependence of runoff on $W^2$—instead of $W$—ensures that the

fractional rate of plant-available water loss is proportional to $W$; relative to a first-order linear kinetics model, this provides a better representation of faster relative plant-available water depletion following high precipitation events, followed by slower losses during lower precipitation timespans (e.g. Matteucci et al., 2015) and serves a functional approximation of both storage-excess and infiltration-excess runoff generation mechanisms in most cases. Following previous results from land surface model development experiments (e.g. Liang et al., 1994; Lawrence et al., 2011), we assume that net runoff inputs from adjacent pixels

are a negligible term in the lumped grid-scale water budget at 4°×5° spatial resolution. By construction, $R_t$ values predicted at $W_t > \frac{1}{a\Delta t}$ are unphysically high ($W_t - R_t\Delta t < 0$), while loss rates at $W_t > \frac{1}{2a\Delta t}$ produce implausibly low residual storage ($W_t - R_t\Delta t$) values. Therefore, in the eventuality of $W_t > \frac{1}{2a\Delta t}$, we calculate runoff as $R_t = W_t - \frac{1}{2a\Delta t}$, effectively represent a storage-excess overflow mechanism by introducing a transition between a state-dependent regime to a direct runoff regime.

We apply a linear scaling on GPP with respect to the plant-available water, where

$$GPP_t = GPP_{\max\,(t)}\,\max\left(1, \frac{W_t}{\omega}\right) \qquad\text{(B3),}$$

where $\omega$ represents the plant-available water stress threshold; Eq. B3 effectively imposes a stress factor on GPP spanning

between 0 and 1, and offers a simplified representation of the integrated effects of leaf-soil water potential differences and their impact on canopy conductance; Evapotranspiration at time $t$ is derived as



$$ET_t = GPP_t \frac{VPD_t}{v_e} \qquad (B4),$$

where $v_e$ is the inherent water use efficiency (Beer et al., 2009) and *VPD* is the vapor pressure deficit derived from ERA-interim monthly reanalysis datasets. Equations B1-B4 amount to a plant-water feedback parameterization, and together represent a reduced complexity version of the DALEC water module implemented by Spadavecchia et al., (2011). All parameters involved in the above-mentioned parameterization—namely $\alpha$, $v_e$, $\omega$ and $W_0$—are optimized along with other DALEC2a parameters in CARDAMOM; the prior ranges are described in Table S1.


**B2. Heterotrophic respiration**

We parameterize the meteorological dependence of heterotrophic respiration, $\rho$ at time *t* as follows:

$$\rho_t = e^{\theta(T_t - \overline{T})} \left( \left( \frac{P_t}{\overline{P}} - 1 \right) s_p + 1 \right) \qquad (B5),$$

where *T* and *P* represent the monthly temperature and precipitation vectors. We chose to use *P* as a driver for heterotrophic respiration sensitivity to moisture, given that (a) the majority of heterotrophic respiration is expected to occur in the near-surface soil layer, and (b) near-surface soil moisture strongly covaries with *P*—rather than water storage—at monthly

timescales. Previous versions of DALEC solely parameterized $\rho_t$ as a function of temperature (e.g. Bloom et al., 2016 and references therein); effectively, the formulation in Eq. B5 induces a joint sensitivity to relative changes in both temperature and near-surface moisture. The prior ranges for the respiration temperature and precipitation sensitivity parameters ($\theta$ and $s_p$) are reported in Table S1.

**Appendix C: Sensitivity of lagged effects to individual ecosystem states**

In the DALEC2a representation of the ecosystem C balance, the state vector $\boldsymbol{x}_a$ consists of the C and $H_2O$ pool values at the start of year *a*. To diagnose the sensitivity of 2010-2015 lagged effects on the variability of ecosystem states, we conduct a sensitivity analysis to explicitly quantify the impact of individual ecosystem state anomalies—relative to their 2010-2015 mean

values—on the variability of $\delta \boldsymbol{x}^{LAG}$ throughout 2010-2015. To do this, we define the anomaly of the *n*th individual state in year *a* as the sum of finite differences relative to the mean state:

$$\boldsymbol{x}_a = \overline{\boldsymbol{x}} + \sum_{n=1}^{N} [\boldsymbol{x}_{a(n)} - \overline{\boldsymbol{x}}] \qquad (C1),$$





where $\overline{x}$ is an $N$-element vector of the mean 2010-2015 states; $N$ is the number of model state variables; $x_{a(n)}$ is an $N$-element vector of ecosystem states, where for the $i$th element $x_{a(n)}(i) = x_a(i)$ for i = n, and $x_{a(n)}(i) = \overline{x}(i)$ for i≠n. Based on Eq. 11 and Eq.14, we can derive the state change under a climatological mean forcing of each term in Eq. C1, and therefore

$$\delta x_a^{LAG} = \delta\overline{x}^{LAG} + \sum_{n=1}^{N}[\delta x_{a(n)}^{LAG} - \delta\overline{x}^{LAG}] + I_a \qquad (C2);$$

$I_a$ collectively accounts for the unaccounted contribution of higher-order interactions between individual pool anomalies $[x_{a(n)} - \overline{x}]$ on $\delta x_a^{LAG}$. As outlined in section 2.4, the "$\delta x$" terms in Equations C2 can be mapped onto any DALEC2a flux variable; specifically, $NBE_a^{LAG}$ can be defined as the sum of lagged effect $NBE$ components attributable to $\delta x_{a(n)}^{LAG}$ and $\delta\overline{x}^{LAG}$ as follows:

$$NBE_a^{LAG} = \overline{NBE}^{LAG} + \sum_{n=1}^{N}[NBE_{a(n)}^{LAG} - \overline{NBE}^{LAG}] + I_a \qquad (C3);$$

$\overline{NBE}^{LAG}$ and $\overline{NBE}_{a(n)}^{LAG}$ can be directly calculated from $D(\overline{x}, M', p)$ and $D(x_{a(n)}, M_a, p)$, respectively. More succinctly, we summarize Eq. B3 as:

$$NBE_a^{LAG} = \overline{NBE}^{LAG} + \sum_{n=1}^{N} \delta NBE_{a(n)}^{LAG} + I_a \qquad (C4),$$

where $\delta NBE_{a(n)}^{LAG}$ represents the lagged effect anomaly attributable solely to the initial condition anomaly in ecosystem state $n$. By applying the "Δ" operator (Eq. 21) on Eq. C3, eq. C4 can alternatively be expressed as:

$$\Delta NBE_a^{LAG} = \sum_{n=1}^{N} \Delta NBE_{a(n)}^{LAG} + \Delta I_a \qquad (C5).$$

Effectively, the lagged effect partitioning formulation outlined in Eq. C5 allows us to quantitatively diagnose the $NBE$ lagged effect dependence on the inter-annual dynamics of individual C and $H_2O$ states depicted in Figure 1.

**Acknowledgments**

Part of this work was carried out at the Jet Propulsion Laboratory, California Institute of Technology, under a contract with the National Aeronautics and Space Administration (NASA), supported by NASA Earth Sciences grant (no. NNH16ZDA001N-IDS). Part of this study was funded as a component of NERC's support of the National Centre for Earth Observation. SSS and AGK were also supported by NASA through the Earth Science program. We are thankful for feedback




from M. Keller and M. Longo. The NCAR MOPITT project is supported by the National Aeronautics and Space Administration (NASA) Earth Observing System (EOS) Program. The MOPITT team acknowledges the contributions of COMDEV and ABB BOMEM with support from the Canadian Space Agency (CSA), the Natural Sciences and Engineering Research Council (NSERC) and Environment Canada.

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

**Figures**


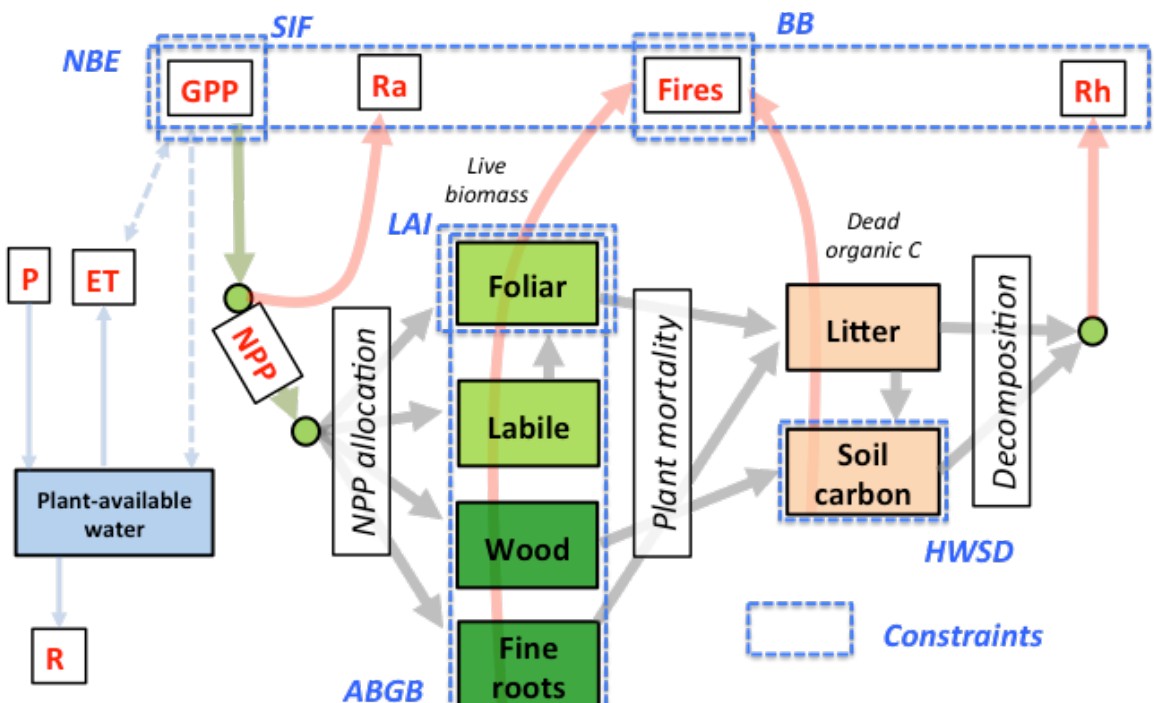

**Figure 1.** Schematic of the CARbon DAta-MOdel fraMework (CARDAMOM) Bayesian model-data fusion approach: the DALEC2a model (described in section 2.1) represents the ecosystem C and plant-available water balance; the dashed blue
boxes denote the observational constraints used in this study (see Table 1 for details). CARDAMOM is implemented at a 4°×5° resolution across the tropics (30°S – 30°N). Within each 4°×5° grid cell, DALEC2a model parameters and initial ecosystem states are optimized using an adaptive Metropolis-Hastings Markov Chain Monte Carlo algorithm.






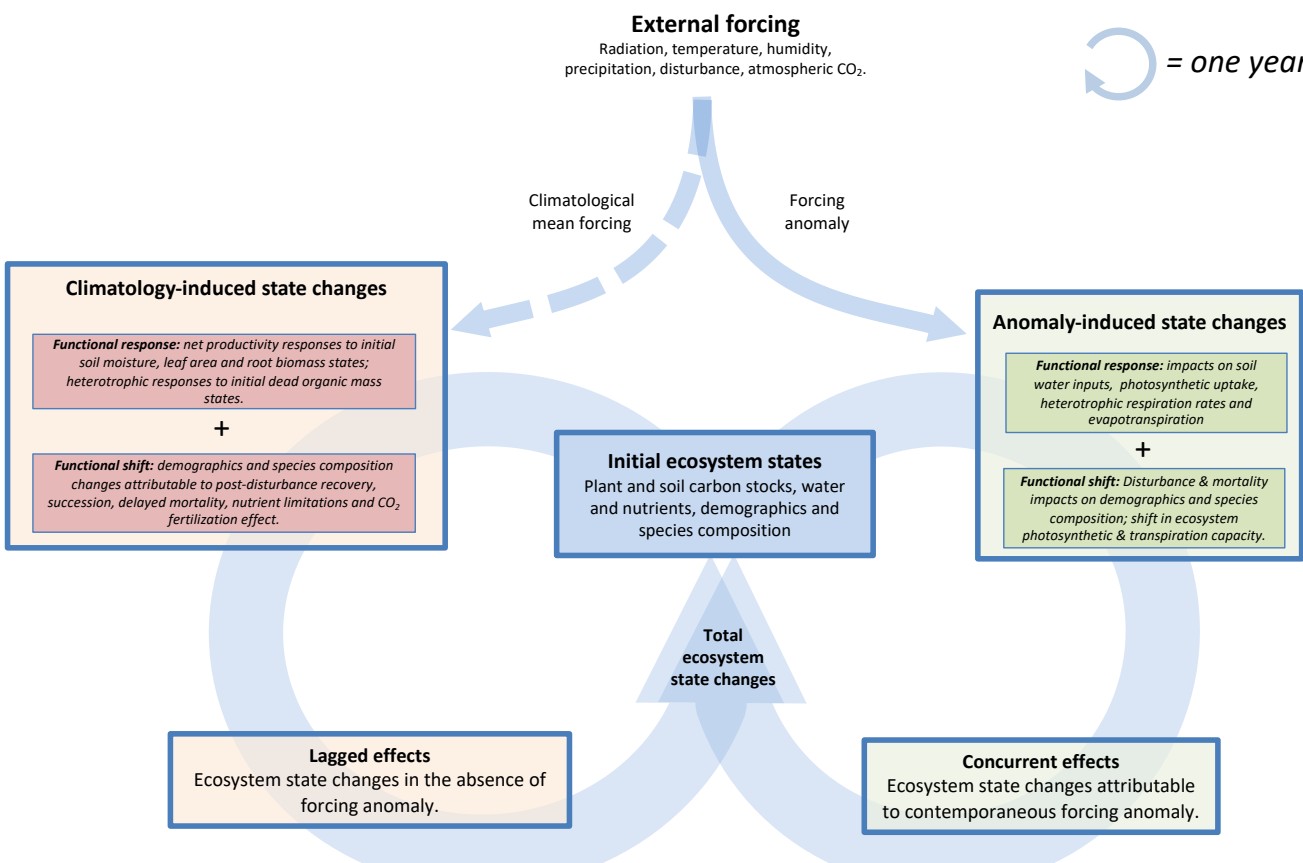

**Figure 2.** Conceptual figure denoting annual ecosystem states changes attributable to concurrent and lagged effects. Throughout a one-year cycle (circular arrows), lagged effects amount to the sum of ecosystem state changes induced by a reference climatological mean forcing, and concurrent effects amount ecosystem state changes solely attributable to a contemporaneous forcing anomaly. The total state changes resulting from both concurrent and lagged effects will in turn determine the next year's initial ecosystem states.




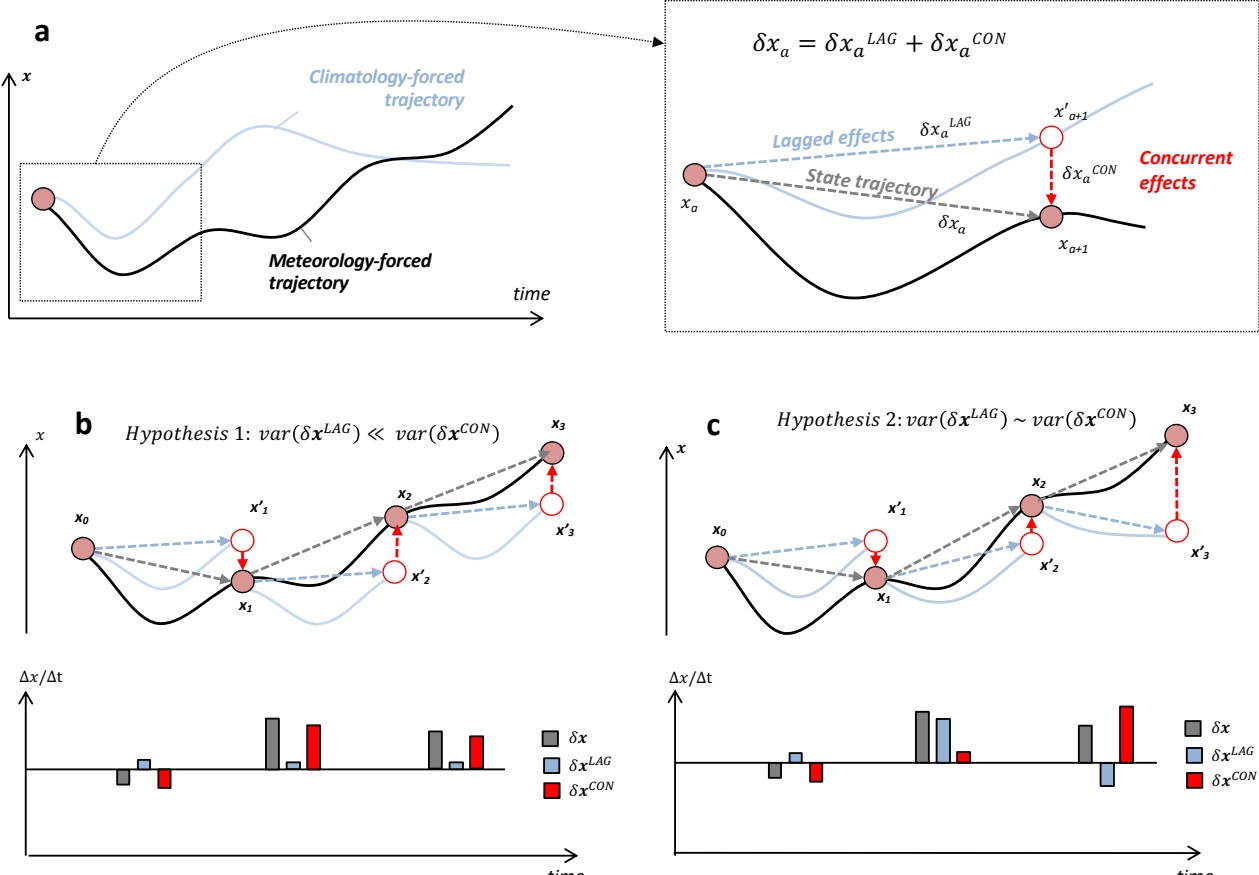

**Figure 3.** (**a**) Schematic of meteorology-forced trajectory of ecosystem state $x$ (solid black line), and trajectory of $x$ under a climatological mean forcing (light blue solid line). Inset: state trajectory $x_a \to x_{a+1}$ ($\delta x_a$), decomposed as the sum of climatology-induced lagged effect vector $x_a \to x'_{a+1}$ ($\delta x_a^{LAG}$) and anomaly-induced concurrent effect vector $x'_{a+1} \to x_{a+1}$ ($\delta x_a^{CON}$). (**b**) Hypothetical scenario depicting approximately time-invariant annual lagged effects $\delta x^{LAG}$ (blue dashed arrows), in reference to changes transient states $x_0, x_1, x_2$, etc.; the temporal changes in $x$ for each time interval, $\delta x$ and $\delta x^{LAG}$ and $\delta x^{CON}$ are shown in the underlying bar chart. In this scenario, $\delta x^{LAG}$ is relatively constant and its variability (denoted as "var()" in schematic equation) is negligible relative to $\delta x^{CON}$. (**c**) Hypothetical scenario depicting time-varying annual lagged effects $\delta x^{LAG}$, in reference to transient states $x_0, x_1, x_2$, etc.; in this scenario, the variability of $\delta x^{CON}$ is comparable to the variability of $\delta x^{LAG}$.





1210

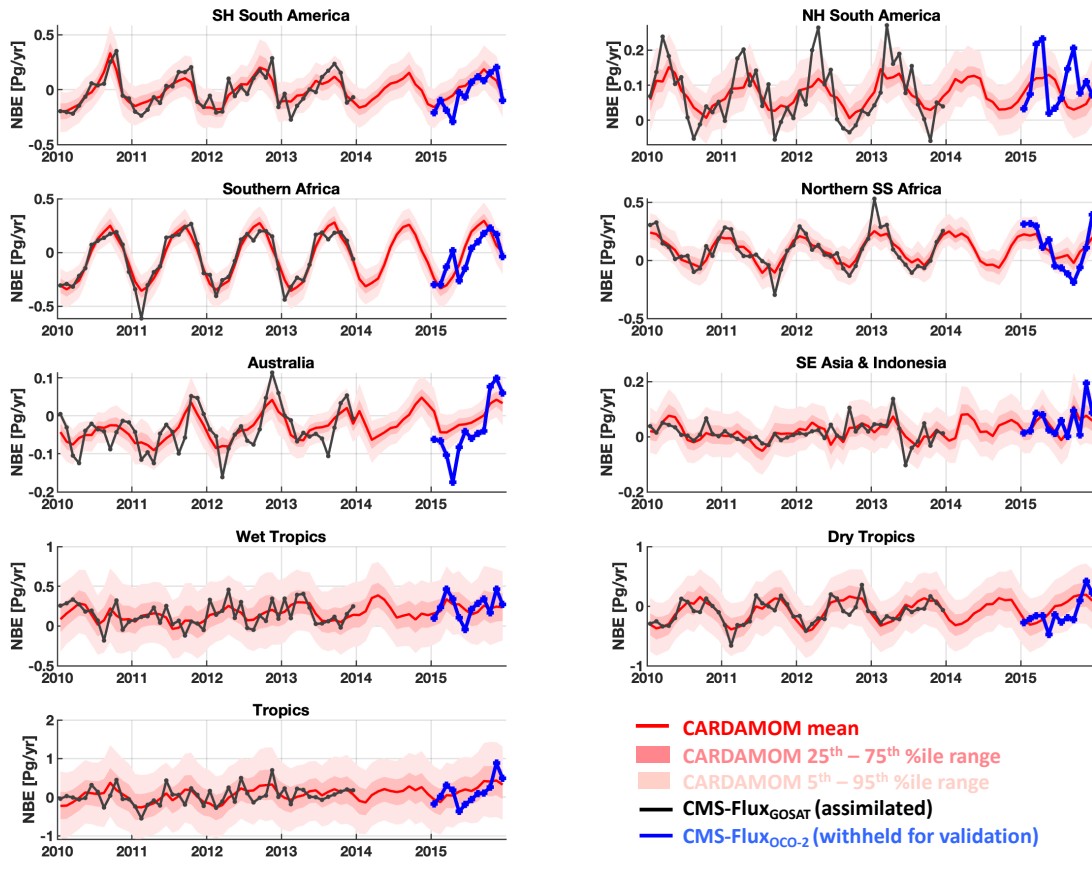

**Figure 4.** CARDAMOM monthly analyses of 2010-2015 median NBE (red line) and associated uncertainty intervals (25th-75th percentiles in dark pink, and 5th – 95th percentiles in light pink). The analyses were constrained by CMS-Flux GOSAT-derived top down fluxes (Liu et al., 2018) for the 2010-2013 period; CMS-Flux OCO-2 derived 2015 NBE fluxes were withheld for validation. The geographical definitions for each region are shown in Figure A1.





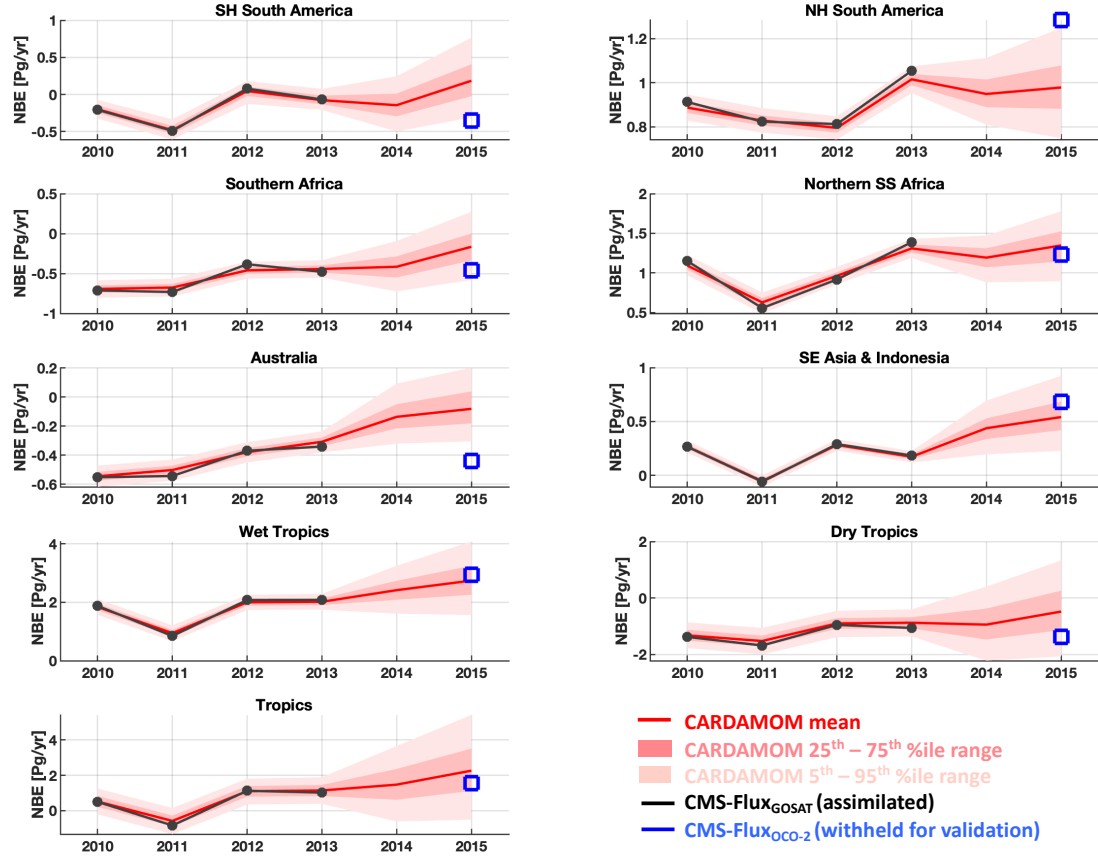

**Figure 5.** CARDAMOM yearly analyses of 2010-2015 NBE (red line) and associated uncertainty intervals (25th-75th percentiles in dark pink, and 5th – 95th percentiles in light pink). The analyses were constrained by CMS-Flux GOSAT-derived top down fluxes (Liu et al., 2018) for the 2010-2013 period. CMS-Flux OCO-2 derived 2015 NBE fluxes were withheld for validation. The geographical definitions for each region are shown in Figure A1.





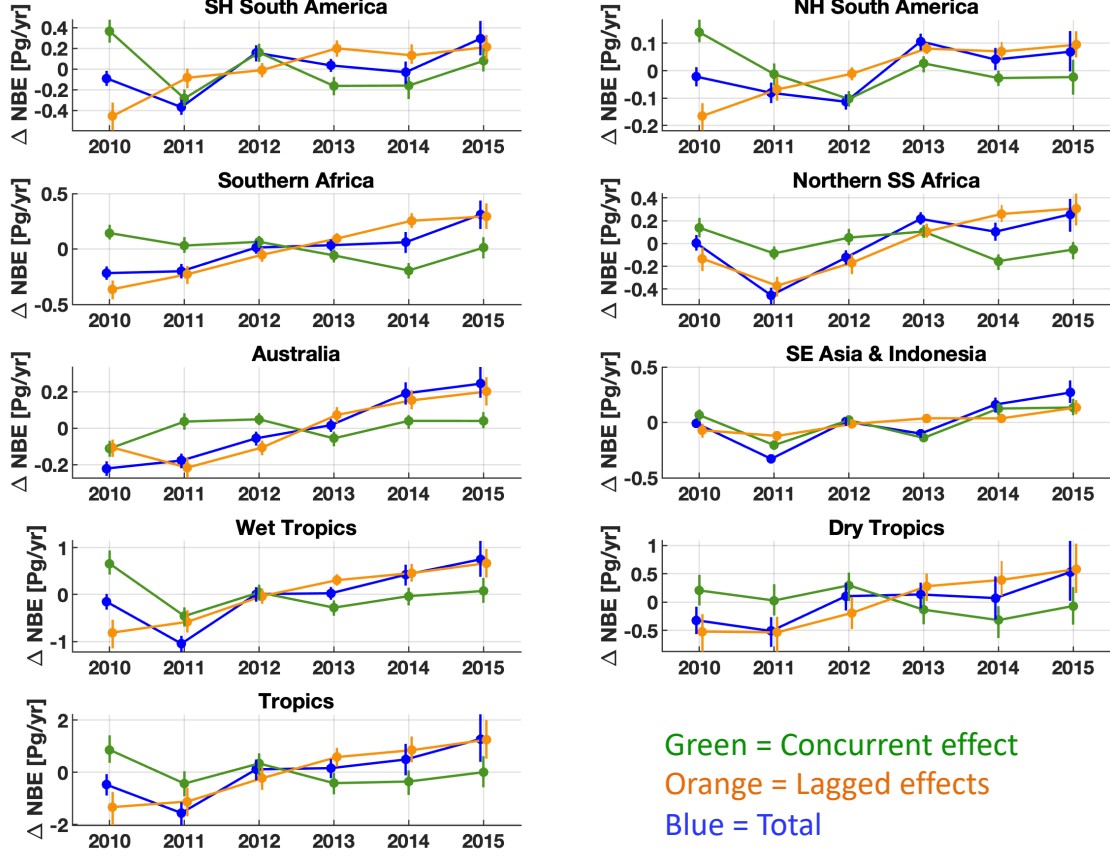

**Figure 6.** Regional and pan-tropical median annual Δ*NBE* (blue line) and its attribution to concurrent effects (Δ*NBE*^CON, green line) and lagged effect (Δ*NBE*^LAG, orange line) components. The geographical definitions for each region are shown in Figure A1. Error bars denote the 25th – 75th percentile uncertainty estimates for each flux anomaly.





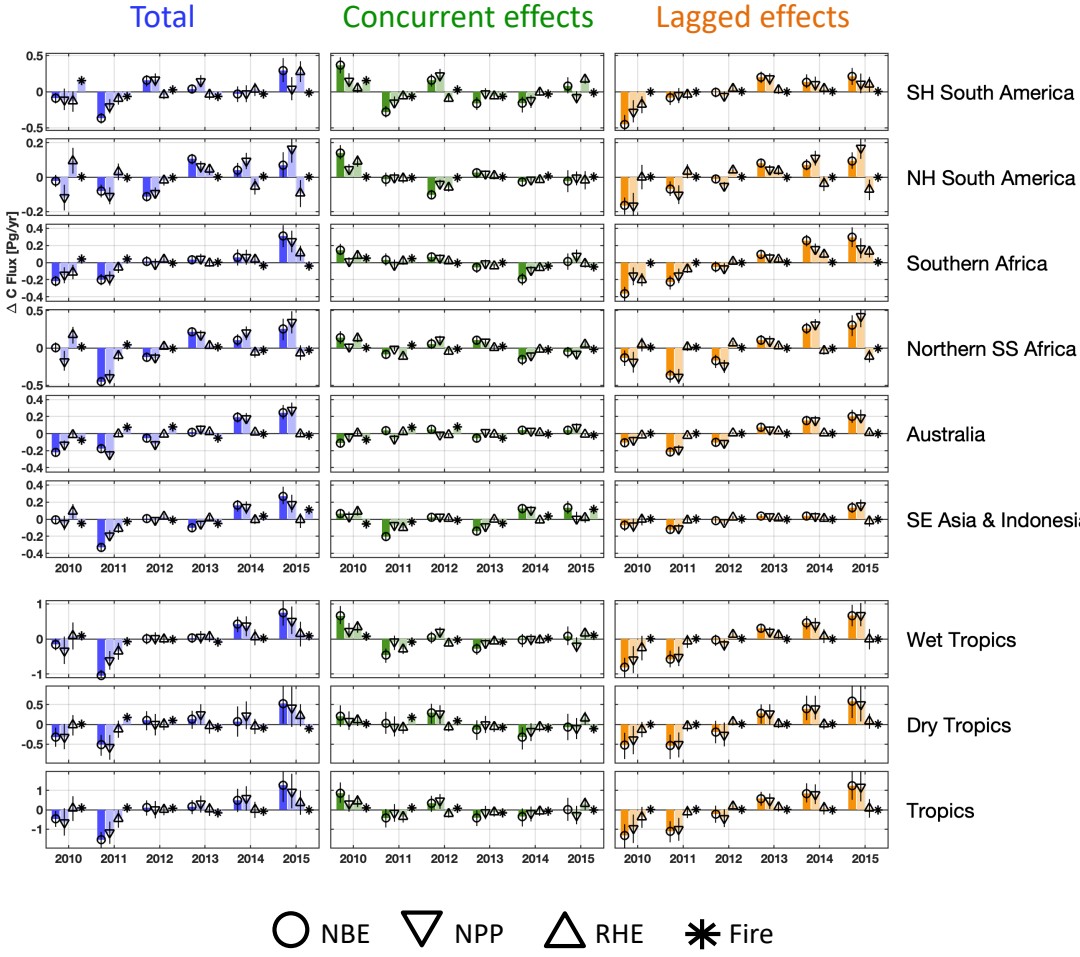

**Figure 7.** Attribution of total NBE anomaly (ΔNBE), concurrent effect ΔNBE ($\Delta NBE^{CON}$ middle column) and lagged effect ΔNBE ($\Delta NBE^{LAG}$ right column) to corresponding annual net primary productivity (NPP), heterotrophic respiration (RHE) and fire components. NPP anomaly signs were reversed such that all anomalies are represented as positive for net land-to-atmosphere C flux. Error bars denote the 25th – 75th percentile uncertainty estimates for each flux anomaly.



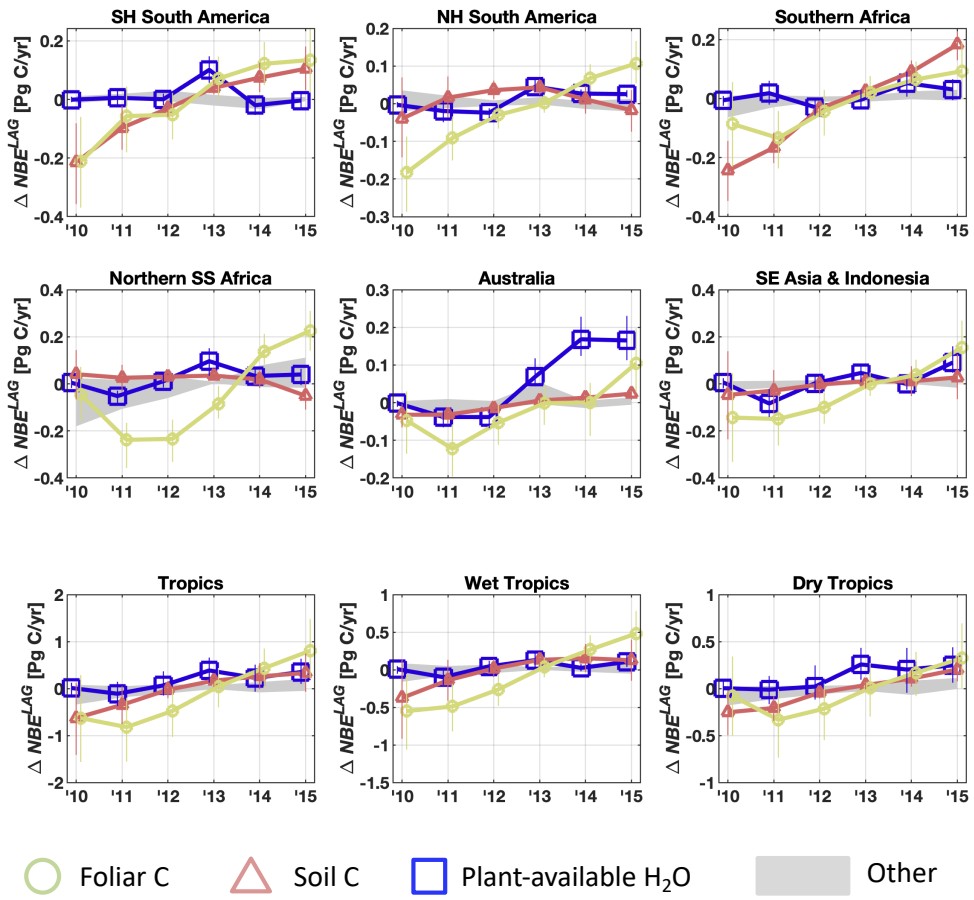

| ○ Foliar C | △ Soil C | □ Plant-available H$_2$O | ▨ Other |

1240

**Figure 8:** Attribution of 2010-2015 annual regional and pan-tropical NBE lagged effect estimates ($\Delta NBE^{LAG}$) to individual ecosystem state anomalies (i.e. the lagged effect in year $a$ solely attributable to anomaly in ecosystem state $n$, $\Delta NBE^{LAG}_{a(n)}$, see Eq. 22). In addition to soil C (dark pink triangles), foliar C (green circles), plant-available H$_2$O (blue squares), the grey areas (labelled as "Other" in the figure legend) denote the collective range of $\Delta NBE^{LAG}$ anomalies attributable to labile, wood, root

1245 and litter C. The sum of annual state-specific $\Delta NBE^{LAG}$ values is approximately equal to the $\Delta NBE^{LAG}$ (see Figure S3). Error bars denote the 25$^{th}$ – 75$^{th}$ percentile uncertainty estimates for each flux anomaly



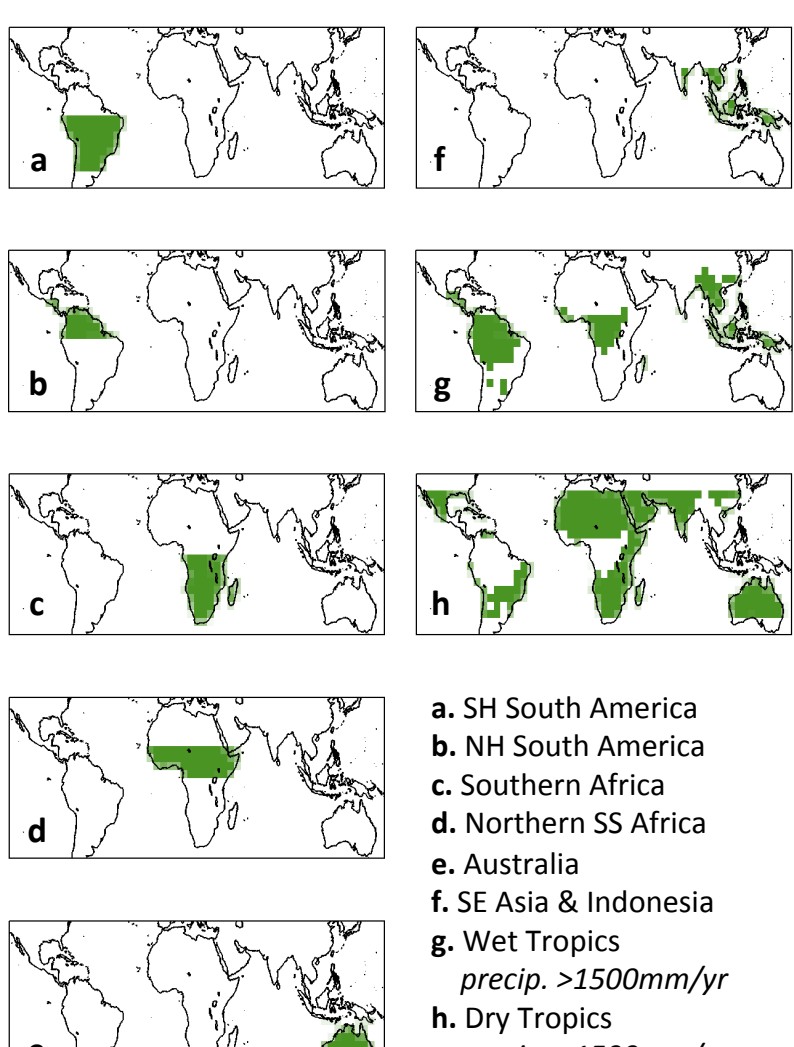

**a.** SH South America
**b.** NH South America
**c.** Southern Africa
**d.** Northern SS Africa
**e.** Australia
**f.** SE Asia & Indonesia
**g.** Wet Tropics
*precip. >1500mm/yr*
**h.** Dry Tropics
*precip. <1500mm/yr*

**Figure A1.** Regional masks used in this study. The 1500mm/yr precipitation thresholds were based on the ERA-interim mean annual precipitation rates throughout the 2010-2015 study period.

250

255



**Tables**

**Table 1.** Observational constraints assimilated into the 4°×5° CARDAMOM simulation.

| Observation | Dataset description | Uncertainty[1] |
|---|---|---|
| Leaf area index (LAI) | MODIS LAI retrievals[2]. | ±log(1.2) |
| Soil organic matter (SOM) | Soil C inventory (Hiederer & Kochy, 2011) | ±log(1.5) |
| Above- and below-ground biomass (ABGB)[5] | GLAS-informed biomass map (Saatchi et al., 2011) | ≥ ±log(1.5)[4] |
| Solar-induced Fluorescence (SIF) | 2010-15 GOSAT retrievals of fluorescence (Frankenberg et al., 2011)[5] | ±log(2) |
| Fire C emissions (BB) | 2010-15 4°×5° inverse estimates of fire C emissions (Worden et al., 2017, Bowman et al., 2017). | ±20% |
| Net Biospheric exchange (NBE) | GOSAT $CO_2$ derived 4°×5° inverse estimates of terrestrial NBE (Liu et al., 2018). | Seasonal = ±2gC/m$^2$/d  Annual= ±0.02gC/m2/d |

[1]Uncertainties denoted as ±log() indicate log-transformed model and observed quantities (i.e. *m* and *o* in Eq. 4).

[1]Only mean 2010-2015 LAI is assimilated into CARDAMOM, in order to mitigate the influence of seasonal LAI retrieval biases (Bi et al., 2015).

[3]The ABGB estimate is applied as a constrain on the sum of all CARDAMOM live biomass pools (Figure 1).

[3]see Bloom et al., 2016 for details.

[5]Time-resoved SIF is assimilated as a relative constraint on the temporal variability of GPP (section 2.3)





**Table 2.** CARDAMOM NBE evaluation against assimilated and predicted NBE

| | Monthly RMSE[a] (Pearson's r) | | Annual RMSE[a,b] (Pearson's r) | |
|---|---|---|---|---|
| | Assimilated NBE (2010-2013) | Predicted NBE (2015) | Assimilated NBE (2010-2013) | Predicted NBE (2015) |
| SH South America | 0.08 (0.86[*]) | 0.10 (0.84[*]) | 0.02 (1.00[*]) | 0.53 ( - ) |
| NH South America | 0.06 (0.75[*]) | 0.09 (-0.11) | 0.02 (0.99[*]) | 0.31 ( - ) |
| Southern Africa | 0.07 (0.94[*]) | 0.14 (0.78[*]) | 0.05 (0.96[*]) | 0.29 ( - ) |
| Northern SS Africa | 0.08 (0.88[*]) | 0.11 (0.94[*]) | 0.06 (1.00[*]) | 0.11 ( - ) |
| Australia | 0.04 (0.73[*]) | 0.06 (0.87[*]) | 0.03 (0.98[*]) | 0.36 ( - ) |
| SE Asia & Indonesia | 0.03 (0.57[*]) | 0.05 (0.58) | 0.01 (1.00[*]) | 0.14 ( - ) |
| Wet Tropics | 0.20 (0.51[*]) | 0.28 (0.51) | 0.14 (1.00[*]) | 0.71 ( - ) |
| Dry Tropics | 0.11 (0.65[*]) | 0.13 (0.51) | 0.07 (1.00[*]) | 0.19 ( - ) |
| Tropics | 0.12 (0.80[*]) | 0.21 (0.55) | 0.13 (0.98[*]) | 0.90 ( - ) |

290

[a]RMSE units are PgC/yr.

[b]Prediction RMSE values are equivalent to absolute errors, since only one error value is considered.

[*]Correlation p-value<0.05

295

300

305

310



**Table 3.** 2010-2015 regional $\Delta NBE$ IAV and corresponding contributions of concurrent effects ($\Delta NBE^{CON}$) and lagged effects ($\Delta NBE^{LAG}$); IAV values are represented here as standard deviations of annual 2010-2015 NBE values; bracketed values represent the Pearson's correlation coefficients between total NBE and concurrent and lagged effect IAV. The regional masks are depicted in Figure A1.

| | $\Delta NBE$ IAV [Pg C/yr] | $\Delta NBE^{CON}$ IAV [as % of $\Delta NBE$ IAV] (Pearson's r) | $\Delta NBE^{LAG}$ IAV [as % of $\Delta NBE$ IAV] (Pearson's r) |
|---|---|---|---|
| SH South America | 0.23 | 108%(0.43) | 111%(0.47) |
| NH South America | 0.09 | 93%(0.28) | 118%(0.64) |
| Southern Africa | 0.19 | 60%(-0.42) | 135%(0.91*) |
| Northern SS Africa | 0.26 | 44%(0.18) | 102%(0.90*) |
| Australia | 0.19 | 35%(0.50) | 88%(0.93*) |
| SE Asia & Indonesia | 0.21 | 68%(0.93*) | 43%(0.84*) |
| Wet Tropics | 0.61 | 63%(0.35) | 97%(0.80) |
| Dry Tropics | 0.37 | 61%(-0.30) | 131%(0.89*) |
| Tropics | 0.96 | 54%(0.03) | 112%(0.87*) |



**Table 4.** Concurrent and lagged effect *NBE* attributed to constituent fluxes (net primary production, heterotrophic respiration and fires, abbreviated as *NPP*, *RHE* and *FIR* respectively): IAV values are represented here as the ratio of constituent flux standard deviation to *NBE* standard deviations of annual 2010-2015 *NBE* values; bracketed values corresponds to Pearson's correlation coefficients between constituent flux and *NBE* ("*" denotes p-values < 0.05). The values highlighted in red denote the largest % IAV contribution to $\Delta NBE^{CON}$ and $\Delta NBE^{LAG}$.

| | IAV as % of $\Delta NBE^{CON}$ (Pearson's r) | | | IAV as % of $\Delta NBE^{LAG}$ (Pearson's r) | | |
|---|---|---|---|---|---|---|
| | $\Delta NPP^{CON}$ | $\Delta RHE^{CON}$ | $\Delta FIR^{CON}$ | $\Delta NPP^{LAG}$ | $\Delta RHE^{LAG}$ | $\Delta FIR^{LAG}$ |
| SH South America | 63%(-0.81) | 40%(-0.41) | 34%(0.32) | 66%(-0.98*) | 39%(0.94*) | 0%(0.45) |
| NH South America | 36%(-0.98*) | 63%(-0.06) | 4%(0.51) | 127%(-0.94*) | 45%(-0.38) | 0%(-0.93*) |
| Southern Africa | 52%(-0.67) | 44%(-0.71) | 36%(-0.67) | 55%(-0.98*) | 46%(0.97*) | 2%(0.99*) |
| Northern SS Africa | 74%(-0.76) | 72%(0.61) | 21%(0.56) | 122%(-0.99*) | 25%(-0.74) | 3%(-0.88*) |
| Australia | 74%(-0.30) | 20%(0.10) | 95%(0.92*) | 92%(-0.99*) | 12%(0.70) | 1%(0.40) |
| SE Asia & Indonesia | 50%(-0.86*) | 44%(-0.11) | 46%(-0.39) | 107%(-0.99*) | 17%(-0.16) | 1%(-0.88*) |
| Wet Tropics | 45%(-0.66) | 58%(-0.60) | 21%(0.20) | 86%(-0.98*) | 25%(0.73) | 1%(-0.90*) |
| Dry Tropics | 71%(-0.88*) | 44%(-0.21) | 51%(-0.46) | 91%(-0.99*) | 16%(0.71) | 0%(0.84*) |
| Tropics | 59%(-0.76) | 60%(-0.55) | 20%(0.30) | 88%(-0.99*) | 20%(0.74) | 0%(-0.72) |





**Table 5.** IAV of 2010-2015 regional and pan-tropical NBE lagged effects attributable to annual anomalies in column-denoted ecosystem states (Eq. 22), as % of total NBE lagged effects ($\Delta NBE^{LAG}$) IAV; bracketed values correspond to Pearson's correlation coefficients between single-state NBE lagged effects and total $\Delta NBE^{LAG}$; "*" denotes p-values < 0.05. The values highlighted in red denote the maximum contribution in each region.

| | Labile C | Foliar C | Fine Root C | Wood C | Litter C | Soil C | Plant-av. $H_2O$ |
|---|---|---|---|---|---|---|---|
| SH South America | 2%(0.80) | 46%(0.98*) | 5%(0.08) | 2%(-0.96*) | 7%(0.15) | 43%(0.97*) | 13%(0.34) |
| NH South America | 11%(0.95*) | 82%(0.95*) | 5%(-0.33) | 15%(-0.96*) | 8%(-0.15) | 25%(0.70) | 16%(0.65) |
| Southern Africa | 11%(0.99*) | 31%(0.95*) | 1%(0.13) | 2%(-0.99*) | 7%(0.77) | 46%(0.99*) | 8%(0.62) |
| Northern SS Africa | 39%(0.86*) | 67%(0.91*) | 2%(-0.29) | 5%(-0.89*) | 7%(-0.39) | 9%(-0.36) | 18%(0.75) |
| Australia | 7%(0.99*) | 34%(0.94*) | 2%(-0.91*) | 0%(-0.29) | 8%(0.34) | 11%(0.94*) | 47%(0.94*) |
| SE Asia & Indonesia | 10%(0.85*) | 55%(0.95*) | 4%(-0.45) | 4%(-0.88*) | 8%(-0.43) | 17%(0.88*) | 26%(0.84*) |
| Wet Tropics | 13%(0.98*) | 53%(0.96*) | 2%(0.29) | 7%(-0.98*) | 5%(0.01) | 34%(0.96*) | 8%(0.68) |
| Dry Tropics | 20%(0.98*) | 40%(0.95*) | 3%(-0.66) | 0%(-0.94*) | 6%(-0.60) | 26%(0.96*) | 23%(0.96*) |
| Tropics | 16%(0.98*) | 47%(0.96*) | 2%(-0.31) | 4%(-0.98*) | 4%(-0.35) | 30%(0.96*) | 14%(0.92*) |