# Peer review of "Lagged effects regulate the inter-annual variability of the tropical carbon balance"

_Biogeosciences, 2019_

## Referee Comment (RC1) · Anonymous Referee #1 · 10 Mar 2020

The paper is generally interesting but it presents too much material, the writing is very dense and further clarity is needed to make it a useful contribution to the literature. I have two main points of criticism: 1) The conceptualisation into concurrent and lagged effects is attractive but also confusing. Only after reading the full manuscript, I actually think I now understand why. Basically, the wording is somehow counterintuitive and I never really understood this until I reached Figure 2. For you, lagged effects refer to the mean climatic effects within a given year while concurrent effects refer to the effects of climatic variability. And actually, both refer to the same year. However, from my initial understanding, probably one shared by many natural scientists, I assumed that the lagged effects are "from what has happened in the years before" and this usually

refers to some strong drought or other extreme event that has longer lasting effects than just within the year itself. So "lagged" effects in this logic are driven by extreme events (such as our concurrent effects) but extending several years backwards. I am not sure if these are really fixed definitions but maybe you should clarify this at a point early in the manuscript and possibly change the naming of your concurrent and lagged effects. Maybe to avoid all that confusion, you could choose an example such as the 2010 Amazon drought and show the usefulness of your framework for this year? This would make the analysis more tangible and easier to follow your logic for the reader. Conceptual clarity would also help to clarify another potential source of confusion: To me it seems you integrate many things as "meteorology forced anomalies" even though things like fire are also influenced by many other, non weather-related factors and it is unclear how you account for that. 2) Related to the former point, I was wondering how meaningful is "lagged" in a 5-year dataset... and why only using 5 years if longer time series are available for most of the data you are using to constraint the model? Related to that, I do not understand why the year 2014 seems missing from the dataset?

Minor comments: L86: "likely critical" L110: i generally agree but this resides on the assumption of the model structure being correct L128: is implement the right word here? As far as I understand, you are using CARDAMOM data to constrain DALEC L131-133: Why is it so important to be consistent with the resolution of the GEOS-Chem modelling? The 4*5° boxes are rather large compared to other model evaluation studies happening at 1 to 0.5° grid resolution. L234: "consists of corresponding" unclear wording... L235: Can you really make this assumption of uncorrelated errors? I would assume that data like LAI, SIF etc have correlated errors? Can you back this up using some relevant literature? L299: word missing L616: better "accounts for a considerable part of NBE variability during" Figure 1: I find this figure rather confusing. Maybe the caption can be improved by clearly identifying and explaining all boxes? Figure 4: I do not understand here: Why is only one year withheld for validation? And why not using longer time series of the data to actually allow for a reasonable evaluation across years? Figure 5 also in my view clearly shows that for several regions the

training period looks good but the evaluation datapoint is not really matched (e.g. SH, NH, Australia). This is not really discussed thoroughly in the manuscript. Figure 6: I would replace "total" with "concurrent + lagged" or sth like that. Table 3/4: I wonder if the lack of significance for the "CON" values should be further explained/discussed.

---

## Editor Comment (EC1) · Andreas Ibrom (Editor) · 11 Mar 2020

Dear Reviewer and Authors, thank you very much for your thoughtful contribution to the interactive discussion.

I have two comments, one for you and one for the Authors.

1. Could you please specify, where the manuscripts contains too much material or suggesting parts that could be omitted without substantial loss, if I understand you correctly?

2. I'd like to support your critical point that the studied period of only five years is

short, maybe too short, to study interannual variability and lagged effects. It is possibly also too short to show that the concept is valid and the approach robust. You suggest demonstrating the usefulness of the concept for the case of the 2010 Amazon drought, which would certainly strengthen the work. In addition, I would like to ask for a case study, maybe at a site scale, where the concept of separating concurrent from lagged effects is demonstrated for a longer time period. Meanwhile there are many long-term flux data sets available. Couldn't those be used for the illustration and corroboration of the concept?

Kind regards, Andreas Ibrom

---

## Referee Comment (RC2) · Anonymous Referee #2 · 16 Mar 2020

Bloom et al rely on a data assimilation approach to constrain parameters of a carbon cycle model (DALEC2a) based on which the analyses separates the contribution of meteorological (CON) versus lagged (LAG) effects on the inter-annual variability (IAV) of net ecosystem carbon fluxes (net biosphere exchange, NBE). The study is focused on tropical regions, where lagged effects on IAV of NBE are comparable to instantaneous effects (CON) but dominate the inter-annual change of NBE. The topic is timely and relevant for the carbon cycle and climate research communities and the methods and datasets represent the state-of-the-art in the field.

This is a study with a clear split between a methodologically elegant approach using

model-data assimilation based results to interpret dynamics of the land surface, and a dataset that seems too short to effectively evaluate the approach which may undermine the robustness of the results. Hence, there should be a bit more stringent evaluation of the model output focused on the IAV component. Of course is hard to do this with only 6 years of NBE, but there is a wealth of data in the study that could be explored e.g. the IAV per month across all months; or the IAV in LAI (which spans ~20 years); also the IAV of SIF; or global observation based products of evapotranspiration (which also span at least a couple decades). The evaluation that is done seems to aggregate across large regions, though the temporal domain per gridcell is not reported: how does the model compare to observations at gridcell level too? In Figures 4 and 5, was there a gridcell comparison within region to ensure that the apparent good behavior is not only a result of the aggregation (e.g. by plotting the distribution of the gridcell differences between model and observations)?

As reading the model-data fusion component there are details missing, many of which are omitted "for the sake of brevity". But there should be a bit more details on the cost function determination, structural uncertainties and dimension of the data streams. How is determined the model structural error (mentioned in line 246)? It is an important component of the assimilation exercise itself but is only mentioned that was determined via a "trial and error" approach (L281). Are the values reported in L284-291? But, does it mean sigma_i in equation 4 is always constant? One other aspect to report is how many observations per data stream in the likelihood function are used? I would suggest the use of the appendix or supplements section to be a bit more descriptive about these approaches and results obtained.

Also for understanding, why not assimilating all the years that have observations? Predicting a point in the future is mostly about evaluating the models' temporal predictive capacity, and not really the model's ability to diagnose the LAG/CON causes of NBE IAV based on available observations. From the results presented (Figure 5) this could lead to a better IAV description? This is probably an experiment computationally too

large to compare, but perhaps there is another motivation behind leaving years out.

As I was going through the LAG versus CON concept and the interpretation of the results, I was wondering: if there is a trend in the carbon pools, whether realistic or not, then the $\Delta NBE$ will be mostly attributed to LAG effects. But the trend is a model output with few constraints and one is left wondering if this has a strong effect in the attribution scheme. This could be demonstrated contrasting the trends in stocks (soil and vegetation stocks) against $\Delta NBE^{LAG}$. But, would it be possible to assess if the trends in the stocks themselves are robust (or too high or too low)?

On the contribution of $\Delta NPP$ to $\Delta NBE$: it seems that it should relate to the contribution of $\Delta GPP$ to $\Delta NBE$ as well once NPP is a fixed fraction of GPP. Was there a reason not to evaluate the IAV of GPP or NPP against independent observation-based datasets?

Other points:

* Section 2.4.: is fair to say then that the maximum lagged effect determined by this approach is of one year.

* Table 1: the numbering of the footnotes is wrong.

* L299: missing verb "we a subset" → "we use a subset" ?

* Eq.12 define $\Delta M\_a$

* L396: "in principle" is repeated and unnecessary

* L399: "NBE^{DIR}\_{a}" should be "NBE^{CON}\_{a}"?

* L511: is very interesting that plant-available $H2O$ is seen to contribute to $\Delta NBE^{LAG}$. Would this mean that controls on NPP via soil moisture would dominate instantaneous controls of precipitation or radiation on photosynthesis in tropical ecosystems?

[Figure]

* L536-538: Is unclear how Figure 7 supports this observation. . .

* Eq. B4: very interesting for this version of DALEC to include v_e. Given its importance on the simulation of water controls on GPP, and the role of plant-available water, is there a spatial gradient from wet to dry tropics?

* L275-276: interesting approach, very similar to Desai 2010 (https://dx.doi.org/10.1029/2010JG001423).

Ultimately, I would like to acknowledge that this manuscript shows a great endeavor in model-data fusion approaches wall-to-wall. Is a follow up on previous developments in the CARDAMOM framework that is clearly of value for the scientific community. But the confidence and interpretation of results is dependent on the evaluation efforts, which would lend robustness to the analysis by focusing more on IAV. I hope that the Authors find these comments useful and I look forward for the feedback.

---

## Author Comment (AC1) · 20 May 2020

**Author response to interactive comments on "Lagged effects dominate the inter-annual variability of the 2010–2015 tropical carbon balance".**

**We thank the reviewers and editor for their constructive feedback and suggested corrections. Below we have addressed each individual comment from each reviewer and the editor (reviewer comments are shown in italics). We believe that the following revisions have substantially improved the overall quality of our manuscript.**

*Reviewer 1.*

**[1.1]** *The paper is generally interesting but it presents too much material, the writing is very dense and further clarity is needed to make it a useful contribution to the literature. I have two main points of criticism: 1) The conceptualisation into concurrent and lagged effects is attractive but also confusing. Only after reading the full manuscript, I actually think I now understand why. Basically, the wording is somehow counterintuitive and I never really understood this until I reached Figure 2. For you, lagged effects refer to the mean climatic effects within a given year while concurrent effects refer to the effects of climatic variability. And actually, both refer to the same year. However, from my initial understanding, probably one shared by many natural scientists, I assumed that the lagged effects are "from what has happened in the years before" and this usually refers to some strong drought or other extreme event that has longer lasting effects than just within the year itself. So "lagged" effects in this logic are driven by extreme events (such as our concurrent effects) but extending several years backwards. I am not sure if these are really fixed definitions but maybe you should clarify this at a point early in the manuscript and possibly change the naming of your concurrent and lagged effects.*

**We thank the reviewer for their comments. We also are not aware of any fixed definition for "lagged effects". We note, however, that our "lagged effects" definition is reconcilable and fundamentally consistent with the reviewer's: we now make an explicit distinction between "single-event lagged effects", attributable to a single forcing event, such as a past extreme event (as proposed by the reviewer), and "aggregate lagged effects", which include the cumulative impacts of past extreme events as well as time-lagged processes induced by nominal climatic variability and longer-term succession processes.**

**We have added the following sentence in section 2 to clarify this:**

**"We note a distinction between (i) single-event lagged effect, which represent ecosystem state changes attributable to a single past forcing event (ii) aggregate lagged effects, which represent the sum and interactions between past single-event lagged effects. For example, single-event lagged effects might include the ecosystem state changes attributable to a single drought or disturbance event, while aggregate lagged effects can include the effects of cumulative droughts impacts, the interactions in between dry and wet year events, and the longer-term succession processes (as described in Figure 2)"**

For clarity, we also state earlier in the manuscript (now at the end of the introduction section, rather than section 2.4) that "We present a framework for expressing the ecosystem state changes in a given year as the sum of (a) "concurrent effects", attributable to concurrent forcing anomalies, and (b) "lagged effects", attributable to the cumulative impacts of past forcings."

Concerning the material quantity, we are confident that the material presented here is necessary for supporting our analysis. However, to better communicate our methodological steps, we have now re-arranged section 2.4 into (a) the conceptual framework of concurrent and lagged effects (now section 2.1) and (b) the *"Dynamical formulation of concurrent and lagged effects"* (now section 2.5).

[1.2] *Maybe to avoid all that confusion, you could choose an example such as the 2010 Amazon drought and show the usefulness of your framework for this year? This would make the analysis more tangible and easier to follow your logic for the reader.*

We agree that an example conveying the concepts presented in our manuscript would be useful for the reader. We now include a supplementary figure demonstrating the successive impact of concurrent forcing anomalies on the variability of NBE lagged effects in the southern-hemisphere South America region.

[1.3] *Conceptual clarity would also help to clarify another potential source of confusion: To me it seems you integrate many things as "meteorology forced anomalies" even though things like fire are also influenced by many other, non weather-related factors and it is unclear how you account for that.*

We agree that our definition of "meteorological forcing anomalies" in fact includes incorporate non-weather factors (namely fires), which we used interchangeably with "forcing anomalies" (as correctly presented in Figure 2). To rectify this, we now replace all instances "meteorological forcing" to "meteorological and disturbance forcing" throughout the text.

[1.4] *2) Related to the former point, I was wondering how meaningful is "lagged" in a 5-year dataset... and why only using 5 years if longer timeseries are available for most of the data you are using to constraint the model?*

We agree that the 2010-2015 period is potentially too short to draw robust conclusions on the inter-annual variability of lagged effects. To address this, we have now extended our analysis to span 15 years (2001-2015). We find that lagged effects remain a prominent component of the extended 2001-2015 time period, albeit (i) concurrent effect NBE IAV is overall on par with lagged effect NBE IAV on a regional basis, and (ii) concurrent effects dominate NBE IAV on a pan-tropical scale (see updated version of Table 3 below). We have revised the manuscript title to "Lagged effects regulate the inter-annual variability of the tropical carbon balance" better reflect this.

**Table 1. Updated version of "Table 3."; values now based on extended 2001-2015 analysis**

| | $\Delta NBE$ IAV [Pg C/yr] | $\Delta NBE^{CON}$ IAV [as % of $\Delta NBE$ IAV] (Pearson's r) | $\Delta NBE^{LAG}$ IAV [as % of $\Delta NBE$ IAV] (Pearson's r) |
|---|---|---|---|
| SH South America | 0.21 | 103%(0.83*) | 58%(0.25) |
| NH South America | 0.08 | 61%(0.16) | 105%(0.83*) |
| Southern Africa | 0.15 | 84%(0.34) | 102%(0.68*) |
| Northern SS Africa | 0.21 | 71%(0.74*) | 62%(0.69*) |
| Australia | 0.13 | 66%(0.57*) | 82%(0.77*) |
| SE Asia & Indonesia | 0.15 | 84%(0.89*) | 46%(0.54*) |
| Wet Tropics | 0.44 | 76%(0.95*) | 36%(0.74*) |
| Dry Tropics | 0.33 | 87%(0.65*) | 78%(0.54*) |
| Tropics | 0.70 | 72%(0.82*) | 57%(0.70*) |

**[1.5]** *Related to that, I do not understand why the year 2014 seems missing from the dataset?*

**The Liu et al., 2018 top-down NBE estimates only span 2010-2013: we now explicitly state this in Table 1 and the methods section.**

*Minor comments:*

**[1.6]** *L86: "likely critical"*

**Correction made**

**[1.7]** *L110: I generally agree but this resides on the assumption of the model structure being correct*

**Agreed. We amended this sentence to clarify that the ability to diagnose concurrent and lagged sensitivities in contingent on the mechanistic accuracy of the C cycle model structure.**

**[1.8]** *L128: Is implement the right word here? As far as I understand, you are using CARDAMOM data to constrain DALEC*

**Yes, we believe so. We now clarify that the CARDAMOM model-data fusion implementation consists of constraining DALEC parameters and initial states with observational constraints data listed in Table 1.**

**[1.9]** *L131-133: Why is it so important to be consistent with the resolution of the GEOS-Chem modelling? The 4\*5◦boxes are rather large compared to other model evaluation studies happening at 1 to 0.5◦grid resolution.*

**We clarify that since DALEC parameters and initial states are independently estimated at each location, the CARDAMOM spatial resolution is fundamentally limited by the coarsest resolution among the observational constraints.**

**[1.10]** *L234: "consists of corresponding" un-clear wording...*

**Now changed to "consists of"**

**[1.11]** *L235: Can you really make this assumption of uncorrelated errors? I would assume that data like LAI, SIF etc have correlated errors? Can you back this up using some relevant literature?*

**We indeed acknowledge the possibility of correlated errors between the datasets, stemming from both 4x5 degree representation errors and systematic errors in commonalities between datasets. We now clarify this in the discussion section of our manuscript as follows:**

**"We also note that the likelihood function (eq. 3) assumes all errors are independent. We note, however, that commonalities in the derived datasets—including systematic representation errors in all datasets, transport errors in the GEOS-Chem derived $CO_2$ and CO emissions, as well as the use of MODIS datasets in both the LAI and the ABGB product—may lead to unrepresented error correlations in the likelihood functions."**

**[1.12]** *L299: word missing*

**Sentence corrected**

**[1.13]** *L616: better "accounts for a considerable part of NBE variability during"*

**Agreed, sentence changed.**

**[1.14]** *Figure 1: I find this figure rather confusing. Maybe the caption can be improved by clearly identifying and explaining all boxes?*

**We have now included (a) an explicit list of observational constraints (blue acronyms), and (b) the fluxes (red acronyms), as part of the figure, and we have adapted the figure caption accordingly.**

**[1.15]** *Figure 4: I do not understand here: Why is only one year withheld for validation? And why not using longer time series of the data to actually allow for a reasonable evaluation across years?*

**We now clarify in section 2 and the Figure 4 caption that satellite-based top-down $CO_2$ NBE estimates are only available for 2010-2013 and 2015 (Liu et al., 2017; Liu et al., 2018). Given the limited timespan of NBE estimates, we chose withhold a single year to evaluate model structural error. Note that we now also include an evaluation of pan-tropical NBE IAV against the 2001-2015 NOAA atmospheric $CO_2$ growth rate estimates (see response to comment 2.1).**

[1.16] *Figure 5 also in my view clearly shows that for several regions the training period looks good but the evaluation datapoint is not really matched (e.g. SH,NH, Australia). This is not really discussed thoroughly in the manuscript.*

**We now elaborate on the significance of these biases in the manuscript (currently reported in lines 553-556 and discussed 702-708), in conjunction with our broader IAV evaluation against independent datasets recommended by reviewer 2 (see response to comment 2.1).**

[1.17] *Figure 6: I would replace "total" with "concurrent + lagged" or sth like that.*

**We have updated figure labels accordingly.**

[1.18] *Table 3/4: I wonder if the lack of significance for the "CON" values should be further explained/discussed.*

**We now find that the majority of regions exhibit significant correlations in the extended 2001-2015 CARDAMOM analysis (see table response to comment 1.4).**

*Reviewer 2.*

[2.1] *Bloom et al rely on a data assimilation approach to constrain parameters of a carbon cycle model (DALEC2a) based on which the analyses separates the contribution of meteorological (CON) versus lagged (LAG) effects on the inter-annual variability (IAV)of net ecosystem carbon fluxes (net biosphere exchange, NBE). The study is focused on tropical regions, where lagged effects on IAV of NBE are comparable to instantaneous effects (CON) but dominate the inter-annual change of NBE. The topic is timely and relevant for the carbon cycle and climate research communities and the methods and datasets represent the state-of-the-art in the field. This is a study with a clear split between a methodologically elegant approach using model-data assimilation based results to interpret dynamics of the land surface, and a dataset that seems too short to effectively evaluate the approach which may undermine the robustness of the results. Hence, there should be a bit more stringent evaluation of the model output focused on the IAV component. Of course is hard to do this with only 6 years of NBE, but there is a wealth of data in the study that could be explored e.g. the IAV per month across all months; or the IAV in LAI (which spans~20 years); also the IAV of SIF; or global observation based products of evapotranspiration (which also span at least a couple decades).*

We thank the reviewer for their comments. In response to comment 1.4 we have now extended the run to span a 15-year time period (2001-2015). As suggested by the reviewer, we now augment the CARDAMOM IAV evaluation by comparing 15-year outputs against independent estimates of GPP, ET and NBE. Specifically, we now include:

(i)    A regional evaluation GPP IAV against two site- satellite-informed global GPP products (Tramontana et al., 2016; Joiner et al., 2018),

(ii)   A regional evaluation of ET against site- and satellite-informed estimates (Tramontana et al., 2016; Mu et al., 2011)

(iii)  A pan-tropical (30N-30S) 2001-2015 NBE IAV against the NOAA atmospheric $CO_2$ growth rate estimates, given the predominant role of the tropical land sink in the global atmospheric $CO_2$ growth rate IAV.

We report the above-mentioned evaluation—as well as assumptions and caveats involved—in sections 2 and 3 of the revised manuscript.

[2.2] *The evaluation that is done seems to aggregate across large regions, though the temporal domain per grid cell is not reported:  how does the model compare to observations at gridcell level too? In Figures 4 and 5, was there a gridcell comparison within region to ensure that the apparent good behavior is not only a result of the aggregation (e.g.  by plotting the distribution of the gridcell differences between model and observations)?*

We have appended Figure S1 to include gridcell-level NBE evaluation against both seasonal variability and inter-annual variability. We note, however, that the basis for aggregation is the limited capacity of the CMS-Flux inversion dataset to accurately resolve fine-scale features, given the fundamental limitations of the information content in the satellite-based surface $CO_2$ flux estimates (Liu et al., 2017 and references therein; we now clarify this in the revised manuscript).

[2.3] *As reading the model-data fusion component there are details missing, many of which are omitted "for the sake of brevity".  But there should be a bit more details on the cost function determination, structural uncertainties and dimension of the data streams.*

We now clarify that the MHMCMC objective function (i.e. the equivalent to exp( - J/2) of a cost-function J) is product p(x) and p(O|x). For completeness, we also document the objective function in the manuscript supplement. See responses to comments 2.4 and 2.5 for the structural uncertainty and dimension of the data streams.

[2.4] *How is determined the model structural error (mentioned in line 246)? It is an important component of the assimilation exercise itself but is only mentioned that was determined via a "trial and error" approach (L281). Are the values reported in L284-291? But, does it mean sigma_i in equation 4 is always constant?*

**We now clarify that observational random errors, systematic errors, and model structural error are jointly implicit in the prescribed uncertainty (sigma) values reported in L284-291 and summarized Table 1; assumed sigma values are indeed constant (either as absolute values or proportional to the magnitude of each observations; see Table 1 for details).**

**[2.5]** *One other aspect to report is how many observations per data stream in the likelihood function are used? I would suggest the use of the appendix or supplements section to be a bit more descriptive about these approaches and results obtained.*

**We have revised Table 1 to explicitly state the number of observations per data stream. In addition, we now include a map of the total number of observations used at each location in the supplement for reference.**

**[2.6]** *Also for understanding, why not assimilating all the years that have observations? Predicting a point in the future is mostly about evaluating the models' temporal predictive capacity, and not really the model's ability to diagnose the LAG/CON causes of NBEIAV based on available observations. From the results presented (Figure 5) this could lead to a better IAV description? This is probably an experiment computationally too large to compare, but perhaps there is another motivation behind leaving years out.*

**We now clarify that our motivation for validating the model's predictive capacity of 2015 NBE is to evaluate the overall mechanistic representation of terrestrial C cycling. Note that we also clarify earlier in the manuscript (in response to comment 1.7) that the mechanistic accuracy of the C cycle model structure is a prerequisite for diagnosing the concurrent and lagged sensitivities of terrestrial ecosystems to external forcing anomalies.**

**[2.7]** *As I was going through the LAG versus CON concept and the interpretation of the results, I was wondering: if there is a trend in the carbon pools, whether realistic or not, then the \DeltaNBE will be mostly attributed to LAG effects. But the trend is a model output with few constraints and one is left wondering if this has a strong effect in the attribution scheme. This could be demonstrated contrasting the trends in stocks (soil and vegetation stocks) against \DeltaNBEˆ{LAG}. But, would it be possible to assess if the trends in the stocks themselves are robust (or too high or too low)?*

**We fully agree with the reviewers assessment: to both (a) quantitatively demonstrate the role of carbon and water states on lagged effect IAV, and (b) evaluate the robustness of the underlying variability and trends in ecosystem states, we support Figure 8 with a supplementary figure showing the relationship between annual states and their corresponding impacts on $\Delta NBE^{LAG}$.**

**[2.8]** *On the contribution of\DeltaNPP to\DeltaNBE: it seems that it should relate to the contribution of \DeltaGPP to\DeltaNBE as well once NPP is a fixed fraction of GPP. Was there a reason not to evaluate the IAV of GPP or NPP against independent observation-based datasets?*

**We now include an evaluation of GPP IAV against two independent datasets (see response to comment 2.1).**

*Other points:*

**[2.9]** *Section 2.4.: is fair to say then that the maximum lagged effect determined by this approach is of one year.*

**Yes, although the conceptual framework can be readily adapted to any timescale. To better convery this, we rephrased the last sentence in section 2.4 as follows: "While in this study we confine our analysis to estimation of concurrent and lagged effects on annual timescales, we note that the conceptual framework presented in Figure 2 can be adapted to diagnose concurrent and lagged ecosystem state changes on any timescale of relevance."**

**[2.10]** *Table 1: the numbering of the footnotes is wrong.*

**Corrections made**

**[2.11]** *L299: missing verb "we a subset"→"we use a subset" ?*

**Correction made**

**[2.12]** *Eq.12 define\deltaM_a*

**Now added "…and \deltaM_a as the corresponding anomaly in year *a,…*"**

**[2.13]** *L396: "in principle" is repeated and unnecessary\**

**Agreed, we removed both instances.**

**[2.14]** *L399: "NBEˆ{DIR}_{a}" should be "NBEˆ{CON}_{a}"?*

**Yes, we now corrected this.**

**[2.15]** *L511:is very interesting that plant-available H2O is seen to contribute to\DeltaNBEˆ{LAG}. Would this mean that controls on NPP via soil moisture would dominate instantaneous controls of precipitation or radiation on photosynthesis in tropical ecosystems?*

**Indeed: for example, $\Delta NPP^{CON}$ and plant-available $H_2O$ contributions to lagged effects exhibit comparable IAV in Australia (~0.2 PgC/yr for 2001-2015). We will further assess the relative importance of plant-available water limitations vs concurrent forcing-induced anomalies in the extended 2001-2015 simulation, and will discuss these appropriately in the revised manuscript.**

**[2.16]** *L536-538: Is unclear how Figure 7 supports this observation...*

**We now clarify that the ΔNPP$^{LAG}$ signs in Figure 7 are reversed to represent all positive anomalies as net land-to-atmosphere fluxes.**

**[2.17]** *Eq. B4: very interesting for this version of DALEC to include v_e. Given its importance on the simulation of water controls on GPP, and the role of plant-available water, is there a spatial gradient from wet to dry tropics?*

**Yes, there is one indeed (albeit with considerable scatter, see Figure below): we find a higher water-use efficiency in the dry tropics (averaging 37 gC hPA/kgH2O at <1500 mm/yr), and lower values in the wet tropics (averaging 28 gC hPA/kgH2O at >1500 mm/yr).**

[Figure]

**[2.18]** *L275-276: interesting approach, very similar to Desai 2010*
*(https://dx.doi.org/10.1029/2010JG001423).*

**Thank you for pointing this out, we now appropriately reference Desai 2010.**

*Ultimately, I would like to acknowledge that this manuscript shows a great endeavor in model-data fusion approaches wall-to-wall. Is a follow up on previous developments in the CARDAMOM framework that is clearly of value for the scientific community. But the confidence and interpretation of results is dependent on the evaluation efforts, which would lend robustness to the analysis by focusing more on IAV. I hope that the Authors find these comments useful and I look forward for the feedback.*

**We thank the reviewers again for their comments and suggestions, we believe these have substantially enhanced the quality of our manuscript.**

*Editor* **[in response to reviewer 1]**

**[3.1: comment addressed to reviewer 1]** *Could you please specify, where the manuscripts contains too much material or suggesting parts that could be omitted without substantial loss, if I understand you correctly?*

**Without removing material, we have re-organized section 2 to better delineate between methodological steps (See response to comment 1.1).**

**[3.2]** *I'd like to support your critical point that the studied period of only five years is short, maybe too short, to study interannual variability and lagged effects. It is possibly also too short to show that the concept is valid and the approach robust. You suggest demonstrating the usefulness of the concept for the case of the 2010 Amazon drought, which would certainly strengthen the work. In addition, I would like to ask for a case study, maybe at a site scale, where the concept of separating concurrent from lagged effects is demonstrated for a longer time period. Meanwhile there are many long-term flux data sets available. Couldn't those be used for the illustration and corroboration of the concept?*

**We agree with both the editor's and reviewer's assessment for the need of a longer timeseries to adequately support our analysis. We have now extended our analysis to span 2001-2015 (see responses to comments 1.4) and have conducted supportive IAV evaluation efforts of the 2001-2015 analysis (see responses to comments 2.1).**

**In response to comment 1.2, we also include a supplementary figure demonstrating the successive impact of concurrent forcing anomalies on NBE lagged effects in the southern-hemisphere South America region.**

**Finally, we acknowledge that site-level datasets can in principle used to diagnose the role of lagged effects on the inter-annual variability of NBE (as mentioned in the last paragraph of section 3). While the implementation of our approach at site level is a substantial effort and beyond the intended scope of our manuscript, we now elaborate on the potential and advantages of implementing our approach at site-level in future efforts.**

**References**

Joiner, J., Yoshida, Y., Zhang, Y., Duveiller, G., Jung, M., Lyapustin, A., ... & Tucker, C. J. (2018). Estimation of terrestrial global gross primary production (GPP) with satellite data-driven models and eddy covariance flux data. *Remote Sensing*, *10*(9), 1346.

Liu, J., Bowman, K. W., Schimel, D. S., Parazoo, N. C., Jiang, Z., Lee, M., ... & O'Dell, C. W. (2017). Contrasting carbon cycle responses of the tropical continents to the 2015–2016 El Niño. *Science*, *358*(6360), eaam5690.

Liu, J., Bowman, K., Parazoo, N. C., Bloom, A. A., Wunch, D., Jiang, Z., ... & Schimel, D. (2018). Detecting drought impact on terrestrial biosphere carbon fluxes over contiguous US with satellite observations. *Environmental Research Letters*, *13*(9), 095003.

Mu, Q., Zhao, M., & Running, S. W. (2011). Improvements to a MODIS global terrestrial evapotranspiration algorithm. *Remote sensing of environment*, *115*(8), 1781-1800.

Tramontana, G., Jung, M., Camps-Valls, G., Ichii, K., Ráduly, B., Reichstein, M., ... & Merbold, L. (2016). Predicting carbon dioxide and energy fluxes across global FLUXNET sites with regression algorithms. *Biogeosciences Discussions*.

---

## Author Response (AR1)

**Author responses and revisions.**

**We thank the reviewers and editor for their constructive feedback and suggested corrections. Below we have addressed each individual comment from each reviewer and the editor (reviewer comments are shown in italics). All manuscript changes are denoted below, and highlighted as 'tracked changes' in the revised manuscript (page and line numbers denote the corresponding changes in the revised manuscript). We believe that the following revisions have substantially improved the overall quality of our manuscript.**

*Reviewer 1.*

**[1.1]** *The paper is generally interesting but it presents too much material, the writing is very dense and further clarity is needed to make it a useful contribution to the literature. I have two main points of criticism: 1) The conceptualisation into concurrent and lagged effects is attractive but also confusing. Only after reading the full manuscript, I actually think I now understand why. Basically, the wording is somehow counterintuitive and I never really understood this until I reached Figure 2. For you, lagged effects refer to the mean climatic effects within a given year while concurrent effects refer to the effects of climatic variability. And actually, both refer to the same year. However, from my initial understanding, probably one shared by many natural scientists, I assumed that the lagged effects are "from what has happened in the years before" and this usually refers to some strong drought or other extreme event that has longer lasting effects than just within the year itself. So "lagged" effects in this logic are driven by extreme events (such as our concurrent effects) but extending several years backwards. I am not sure if these are really fixed definitions but maybe you should clarify this at a point early in the manuscript and possibly change the naming of your concurrent and lagged effects.*

**We thank the reviewer for their comments. We also are not aware of any fixed definition for "lagged effects". We note, however, that our "lagged effects" definition is reconcilable and fundamentally consistent with the reviewer's: we now make an explicit distinction between "single-event lagged effects", attributable to a single forcing event, such as a past extreme event (as proposed by the reviewer), and "aggregate lagged effects", which include the cumulative impacts of past extreme events as well as time-lagged processes induced by nominal climatic variability and longer-term succession processes.**

**We have added the following sentence in section 2 (P6 L9-15) to clarify this:**

**"We note a distinction between (i) single-event lagged effect, which represent ecosystem state changes attributable to a single past forcing event (ii) aggregate lagged effects, which represent the sum and interactions between past single-event lagged effects. For example, single-event lagged effects might include the ecosystem state changes attributable to a single drought or disturbance event, while aggregate lagged effects can include the effects of cumulative droughts impacts, the interactions in between dry and wet year events, and the longer-term succession processes (as described in Figure 1); we henceforth use "lagged effects" to refer to aggregate lagged effects throughout the manuscript."**

For clarity, we also state earlier in the manuscript (now at the end of the introduction section, rather than section 2, P4 L15-17) that "we present a framework for expressing the ecosystem state changes in a given year as the sum of (a) "concurrent effects", attributable to concurrent forcing anomalies, and (b) "lagged effects", attributable to the cumulative impacts of past forcings."

Concerning the material quantity, we are confident that the material presented here is necessary for supporting our analysis. However, to better communicate our methodological steps, we have now re-arranged section 2.4 into (a) the conceptual framework of concurrent and lagged effects (now section 2.1) and (b) the dynamical formulation of concurrent and lagged effects (now section 2.5).

[1.2] *Maybe to avoid all that confusion, you could choose an example such as the 2010 Amazon drought and show the usefulness of your framework for this year? This would make the analysis more tangible and easier to follow your logic for the reader.*

We agree that an example conveying the concepts presented in our manuscript would be useful for the reader. We now include a supplementary figure (Figure S6) demonstrating the successive impact of 2001-2015 concurrent forcing anomalies on the variability of NBE lagged effects in the southern-hemisphere South America region. We refer to this figure and report noteworthy results in section 3.2 (P19 L5-19).

[1.3] *Conceptual clarity would also help to clarify another potential source of confusion: To me it seems you integrate many things as "meteorology forced anomalies" even though things like fire are also influenced by many other, non weather-related factors and it is unclear how you account for that.*

We agree that our definition of "meteorological forcing anomalies" in fact includes incorporate non-weather factors (namely fires), which we used interchangeably with "forcing anomalies" (as correctly presented in Figure 1). To rectify this, we now replace all instances "meteorological forcing" to "meteorological and disturbance forcing" throughout the text.

[1.4] *2) Related to the former point, I was wondering how meaningful is "lagged" in a 5-year dataset... and why only using 5 years if longer timeseries are available for most of the data you are using to constraint the model?*

We agree that the 2010-2015 period is potentially too short to draw robust conclusions on the inter-annual variability of lagged effects. To address this, we have now extended our analysis to span 15 years (2001-2015). We find that lagged effects remain a prominent component of the extended 2001-2015 time period, albeit (i) concurrent effect NBE IAV is overall on par with lagged effect NBE IAV on a regional basis, and (ii) concurrent effects dominate NBE IAV on a pan-tropical scale. We have revised the manuscript title to "Lagged effects regulate the inter-annual variability of the tropical carbon balance" better reflect this.

**In the revised manuscript we have (a) updated Figures 3-8 and Tables 2-5 to report results throughout 2001-2015 period; (b) updated text throughout sections 2 and 3 accordingly, and (c) updated abstract and conclusions to appropriately reflect changes (supporting figures in the supplementary material were also updated accordingly).**

**[1.5]** *Related to that, I do not understand why the year 2014 seems missing from the dataset?*

**The Liu et al., 2018 top-down NBE estimates only span 2010-2013. In the revised manuscript, we now explicitly state the Liu et al., 2018 timespan in section 2.3 (P8 L10) and Table 1.**

*Minor comments:*

**[1.6]** *L86: "likely critical"*

**Correction made (P3 L21)**

**[1.7]** *L110: I generally agree but this resides on the assumption of the model structure being correct*

**Agreed. We amended this sentence (P4 L8-13) to clarify that the ability to diagnose concurrent and lagged sensitivities in contingent on the mechanistic accuracy of the C cycle model structure.**

**[1.8]** *L128: Is implement the right word here? As far as I understand, you are using CARDAMOM data to constrain DALEC*

**Yes, we believe so. We now clarify that the CARDAMOM model-data fusion implementation consists of constraining DALEC parameters and initial states with observational constraints.**

**We now include the following statement in the revised manuscript (P5 L2-5): "In summary, the CARDAMOM model-data fusion framework (Bloom et al., 2016) employs a Bayesian inference approach to constrain model parameters and initial states within the prognostic Data Assimilation Linked Ecosystem Carbon model (DALEC, Williams et al., 2005), based on observation constraints—where and when these are available."**

**[1.9]** *L131-133: Why is it so important to be consistent with the resolution of the GEOS-Chem modelling? The 4\*5◦boxes are rather large compared to other model evaluation studies happening at 1 to 0.5◦grid resolution.*

**We clarify that since DALEC parameters and initial states are independently estimated at each location, the CARDAMOM spatial resolution is fundamentally limited by the coarsest resolution among the observational constraints (P5 L5-7).**

**[1.10]** *L234: "consists of corresponding" un-clear wording…*

**Now changed to "consists of" (P9 L13)**

**[1.11]** *L235: Can you really make this assumption of uncorrelated errors? I would assume that data like LAI, SIF etc have correlated errors? Can you back this up using some relevant literature?*

**We indeed acknowledge the possibility of correlated errors between the datasets, stemming from both 4x5 degree representation errors and systematic errors in commonalities between datasets. We now clarify this in section 3.3 of the revised manuscript (P20 L28-31).**

**[1.12]** *L299: word missing*

**Sentence corrected (P11 L17)**

**[1.13]** *L616: better "accounts for a considerable part of NBE variability during"*

**Sentence has been removed in revised manuscript in response to comment 1.4**

**[1.14]** *Figure 1: I find this figure rather confusing. Maybe the caption can be improved by clearly identifying and explaining all boxes?*

**We now spell out the flux acronyms (boxes with red acronyms) as part of the figure (Figure 2 in the revised manuscript). To avoid duplication, we now state in the figure caption that details on both the observational constraint acronyms and associated details are provided in Table 1.**

**[1.15]** *Figure 4: I do not understand here: Why is only one year withheld for validation? And why not using longer time series of the data to actually allow for a reasonable evaluation across years?*

**We now highlight in section 2.3 (P8 L10, L18-21) that satellite-based top-down CO$_2$ NBE estimates are only available for 2010-2013 and 2015 (Liu et al., 2017; Liu et al., 2018). Given the limited timespan of NBE estimates, we chose withhold a single year (2015) to evaluate model structural error. Note that we now also include an evaluation of pan-tropical NBE IAV against the 2001-2015 NOAA atmospheric CO$_2$ growth rate estimates (see response to comment 2.1).**

**[1.16]** *Figure 5 also in my view clearly shows that for several regions the training period looks good but the evaluation datapoint is not really matched (e.g. SH,NH, Australia). This is not really discussed thoroughly in the manuscript.*

In the updated 2001-2015 analysis, we now find that all regions except NH South America are within the 5th – 95th percentile range (see updated Figure 5; we have updated the text in P15 L16-17 accordingly). We have also adapted the text in section 3.3 (P21 L34 – P22 L4) to specifically discuss the underestimation of 2015 NBE in NH South America.

[1.17] *Figure 6: I would replace "total" with "concurrent + lagged" or sth like that.*

We have updated figure label to state "Blue = Total (concurrent + lagged)"

[1.18] *Table 3/4: I wonder if the lack of significance for the "CON" values should be further explained/discussed.*

We now find that the majority of regions exhibit significant correlations in the extended 2001-2015 CARDAMOM analysis (see updated values in Tables 3 and 4 in revised manuscript).

*Reviewer 2.*

[2.1] *Bloom et al rely on a data assimilation approach to constrain parameters of a carbon cycle model (DALEC2a) based on which the analyses separates the contribution of meteorological (CON) versus lagged (LAG) effects on the inter-annual variability (IAV)of net ecosystem carbon fluxes (net biosphere exchange, NBE). The study is focused on tropical regions, where lagged effects on IAV of NBE are comparable to instantaneous effects (CON) but dominate the inter-annual change of NBE. The topic is timely and relevant for the carbon cycle and climate research communities and the methods and datasets represent the state-of-the-art in the field. This is a study with a clear split between a methodologically elegant approach using model-data assimilation based results to interpret dynamics of the land surface, and a dataset that seems too short to effectively evaluate the approach which may undermine the robustness of the results. Hence, there should be a bit more stringent evaluation of the model output focused on the IAV component. Of course is hard to do this with only 6 years of NBE, but there is a wealth of data in the study that could be explored e.g. the IAV per month across all months; or the IAV in LAI (which spans ~20 years); also the IAV of SIF; or global observation based products of evapotranspiration (which also span at least a couple decades).*

We thank the reviewer for their comments. In response to comment 1.4 we have now extended the run to span a 15-year time period (2001-2015). As suggested by the reviewer, we now augment the CARDAMOM IAV evaluation; specifically, we comparing 2001-2015 CARDAMOM outputs against (i) independent estimates of GPP and ET, (ii) MODIS LAI (given that only mean LAI values are assimilated in CARDAMOM) and, (iii) atmospheric $CO_2$ growth rate, as a proxy for pan-tropical NBE IAV.

For NBE: we have now revised Figure 5 to include an evaluation of CARDAMOM NBE IAV against the NOAA ESRL detrended annual estimates of global atmospheric $CO_2$ growth rate. We assume that the detrended NOAA $CO_2$ growth rate variability predominantly exhibit

inter-annual variations of the tropical C balance; we report the evaluation and state underlying assumptions in section 3.1 (P15 L26 – P16 L2).

**For GPP, ET and LAI:** we now include an evaluation of regional CARDAMOM IAV against two independent GPP estimates (Jung et al., 2020; Joiner et al., 2018), two independent ET estimates (Jung et al., 2019; Mu et al., 2011) and MODIS LAI IAV. We report the noteworthy evaluation outcomes in the main body of the manuscript (P16 L12-32); we include the corresponding datasets details and regional evaluations in the supplementary material (section S2 and Tables S2-S3).

**[2.2]** *The evaluation that is done seems to aggregate across large regions, though the temporal domain per grid cell is not reported:  how does the model compare to observations at gridcell level too? In Figures 4 and 5, was there a gridcell comparison within region to ensure that the apparent good behavior is not only a result of the aggregation (e.g.  by plotting the distribution of the gridcell differences between model and observations)?*

We have appended Figure S2 to include gridcell NBE evaluation against both seasonal variability and inter-annual variability. We note, however, that the basis for aggregation is the limited capacity of the CMS-Flux inversion dataset to accurately resolve fine-scale features, given the fundamental limitations of the information content in the satellite-based surface $CO_2$ flux estimates.

In the revised manuscript (P5 L9-11), we now include the following statement "we chose to focus our evaluation on sub-continental and pan-tropical scales to account for the fundamental spatial resolution limitations of satellite-based surface $CO_2$ flux estimates (Liu et al., 2014, Bowman et al., 2017)." We also report the gridcell level seasonal and inter-annual evaluation in section 3.1 (P15 L21-25).

**[2.3]** *As reading the model-data fusion component there are details missing, many of which are omitted "for the sake of brevity".  But there should be a bit more details on the cost function determination, structural uncertainties and dimension of the data streams.*

We now clarify that the MHMCMC objective function (i.e. the equivalent to exp( - J/2) of a cost-function J) is product p(x) and p(O|x) (P11 L10-11). For completeness, we also document the objective function in the manuscript supplement (section S3 and table S4). See responses to comments 2.4 and 2.5 for the structural uncertainty and dimension of the data streams.

**[2.4]** *How is determined the model structural error (mentioned in line 246)? It is an important component of the assimilation exercise itself but is only mentioned that was determined via a "trial and error" approach (L281). Are the values reported in L284-291? But, does it mean sigma_i in equation 4 is always constant?*

We have added a sentence (P11 L4-6) to clarify that observational random errors, systematic errors, and model structural error are jointly implicit in the prescribed uncertainty (sigma) values (as reported in P10 L30 – P11 L4 and summarized Table 1); assumed sigma values are indeed constant (either as absolute values or proportional to the magnitude of each observations; see Table 1).

[2.5] *One other aspect to report is how many observations per data stream in the likelihood function are used? I would suggest the use of the appendix or supplements section to be a bit more descriptive about these approaches and results obtained.*

We have revised Table 1 to explicitly state the number of observations per data stream. In addition, we now include the corresponding observation coverage maps in the supplement for reference (Figure S1 in the updated supplement).

[2.6] *Also for understanding, why not assimilating all the years that have observations? Predicting a point in the future is mostly about evaluating the models' temporal predictive capacity, and not really the model's ability to diagnose the LAG/CON causes of NBEIAV based on available observations. From the results presented (Figure 5) this could lead to a better IAV description? This is probably an experiment computationally too large to compare, but perhaps there is another motivation behind leaving years out.*

We now clarify that our motivation for validating the model's predictive capacity of 2015 NBE is to evaluate the overall mechanistic representation of terrestrial C cycling. We now clarify this in P8 L18-23 of the revised manuscript.

We also clarify earlier in the manuscript that the mechanistic accuracy of the C cycle model structure is a prerequisite for diagnosing the concurrent and lagged sensitivities of terrestrial ecosystems to external forcing anomalies (see response to comment 1.7)

[2.7] *As I was going through the LAG versus CON concept and the interpretation of the results, I was wondering: if there is a trend in the carbon pools, whether realistic or not, then the \DeltaNBE will be mostly attributed to LAG effects. But the trend is a model output with few constraints and one is left wondering if this has a strong effect in the attribution scheme. This could be demonstrated contrasting the trends in stocks (soil and vegetation stocks) against \DeltaNBEˆ{LAG}. But, would it be possible to assess if the trends in the stocks themselves are robust (or too high or too low)?*

We fully agree with the reviewer's assessment. In the revised manuscript we now include:

1. 2001-2015 relative foliar C, soil C and plant-available $H_2O$ IAV (%) within each panel in Figure 8.
2. A supplementary figure (Figure S5) showing the relative variations of 2001-2015 CARDAMOM states and correlations with corresponding pool-specific $\Delta NBE^{LAG}$ timeseries.

3. An assessment of foliar C, soil C and plant-available $H_2O$ IAVs and trends, and their relationship with pool-specific $\Delta NBE^{LAG}$ timeseries (P19 L19-27).

**[2.8]** *On the contribution of \DeltaNPP to \DeltaNBE: it seems that it should relate to the contribution of \DeltaGPP to \DeltaNBE as well once NPP is a fixed fraction of GPP. Was there a reason not to evaluate the IAV of GPP or NPP against independent observation-based datasets?*

**As recommended by the reviewer, we now include an evaluation of GPP IAV against two independent GPP estimates (see response to comment 2.1).**

*Other points:*

**[2.9]** *Section 2.4.: is fair to say then that the maximum lagged effect determined by this approach is of one year.*

**Yes, although the conceptual framework can be readily adapted to any timescale. To better convey this, we rephrased the last sentence in section 2.1 (P6 L15-17) as follows: "Finally, while in this study we confine our analysis to estimation of concurrent and lagged effects on annual timescales, we note that the conceptual framework presented in Figure 1 can be adapted to diagnose concurrent and lagged ecosystem state changes on any timescale of relevance."**

**[2.10]** *Table 1: the numbering of the footnotes is wrong.*

**Corrections made**

**[2.11]** *L299: missing verb "we a subset"→"we use a subset" ?*

**Correction made**

**[2.12]** *Eq.12 define \deltaM_a*

**Now added "…and \deltaM_a as the corresponding anomaly in year *a*,…" (P12 L3-4)**

**[2.13]** *L396: "in principle" is repeated and unnecessary\**

**Agreed, we removed both instances.**

**[2.14]** *L399: "NBEˆ{DIR}_{a}" should be "NBEˆ{CON}_{a}"?*

**Yes, we now corrected this.**

**[2.15]** *L511:is very interesting that plant-available H2O is seen to contribute to\DeltaNBEˆ{LAG}. Would this mean that controls on NPP via soil moisture would dominate instantaneous controls of precipitation or radiation on photosynthesis in tropical ecosystems?*

**Indeed: for example, $\Delta NPP^{CON}$ and plant-available $H_2O$ contributions to lagged effects exhibit comparable IAV in Australia (~0.2 PgC/yr for 2001-2015). We added the following sentence in the revised manuscript to explicitly highlight the relative importance of both plant-available H2O and foliar C on NPP (P19 L16-19): "We also find that the sum of regional foliar C and plant-available H2O impacts on ΔNBELAG (Figure 8) are approximately equivalent to ΔNPPLAG (Figure 7); in turn, the considerable contributions of both ΔNPPLAG¬ and ΔNPPCON across tropical ecosystems indicates that both climatic variability and initial ecosystem states are substantial contributors to tropical ΔNPP IAV."**

**[2.16]** *L536-538: Is unclear how Figure 7 supports this observation...*

**This sentence was removed in the revised manuscript.**

**[2.17]** *Eq. B4: very interesting for this version of DALEC to include v_e. Given its importance on the simulation of water controls on GPP, and the role of plant-available water, is there a spatial gradient from wet to dry tropics?*

**Yes, there is one indeed (albeit with considerable scatter, see Figure below): we find a higher ʋ_e (inherent water-use efficiency) in the dry tropics (averaging 37 gC hPA/kgH2O at <1500 mm/yr), and lower values in the wet tropics (averaging 28 gC hPA/kgH2O at >1500 mm/yr).**

**Ultimately, we chose not to include this figure in our revised manuscript; however, we now discuss the potential sensitivity of CARDAMOM ET and plant-available water IAV to the assumed GPP:ET relationship (P21 L22-26).**

[Figure]

**[2.18]** *L275-276: interesting approach, very similar to Desai 2010 (https://dx.doi.org/10.1029/2010JG001423).*

**Thank you for pointing this out, we now appropriately reference Desai 2010 (P10 L19-20)**

*Ultimately, I would like to acknowledge that this manuscript shows a great endeavor in model-data fusion approaches wall-to-wall. Is a follow up on previous developments in the CARDAMOM framework that is clearly of value for the scientific community. But the confidence and interpretation of results is dependent on the evaluation efforts, which would lend robustness to the analysis by focusing more on IAV. I hope that the Authors find these comments useful and I look forward for the feedback.*

**We thank the reviewers again for their comments and suggestions, we believe these have substantially enhanced the quality of our manuscript.**

*Editor* **[in response to reviewer 1]**

**[3.1: editor comment addressed to reviewer 1]** *Could you please specify, where the manuscripts contains too much material or suggesting parts that could be omitted without substantial loss, if I understand you correctly?*

**Without removing material, we have re-organized section 2 to better delineate between methodological steps (see response to comment 1.1).**

**[3.2]** *I'd like to support your critical point that the studied period of only five years is short, maybe too short, to study interannual variability and lagged effects. It is possibly also too short to show that the concept is valid and the approach robust. You suggest demonstrating the usefulness of the concept for the case of the 2010 Amazon drought, which would certainly strengthen the work. In addition, I would like to ask for a case study, maybe at a site scale, where the concept of separating concurrent from lagged effects is demonstrated for a longer time period. Meanwhile there are many long-term flux data sets available. Couldn't those be used for the illustration and corroboration of the concept?*

**We agree with both the editor's and reviewer's assessment for the need of a longer timeseries to adequately support our analysis. We have now extended our analysis to span 2001-2015 (see responses to comments 1.4) and have conducted supportive IAV evaluation efforts of the 2001-2015 analysis (see responses to comments 2.1).**

**In response to comment 1.2, we also include a supplementary figure (Figure S6) demonstrating the successive impact of concurrent forcing anomalies on NBE lagged effects in the southern-hemisphere South America region; as outlined in our response to comment 1.2, we refer to this figure and report noteworthy results in section 3.2 (P19 L5-19).**

**Finally, we acknowledge that site-level datasets can in principle used to diagnose the role of lagged effects on the inter-annual variability of NBE. While the implementation of our approach at site level is a substantial effort and beyond the intended scope of our manuscript, we have now reworded the relevant text in section 3.3 (P22 L9-13) to highlight the advantages of implementing our approach at site-level in future efforts.**

**Additional changes**

- **Section 3.2: given the 2001-2015 yields overall larger uncertainties in concurrent, lagged at total NBE anomalies (Figures 6 and 7), we now (a) report the confounding source of uncertainty, and (b) explicitly quantify the uncertainty of lagged effect contributions to NBE IAV, in order to verify that these are significant irrespective of annual NBE uncertainties (P17 L29 – P18 L2).**
- **Figure 6: We now present results in bar chart format; we found the updated layout improved the legibility of the full 2001-2015 sequence of concurrent, lagged at total NBE anomalies.**

- Figure 7: we now (i) display only primary productivity, respiration and fires in order to avoid repetition (as NBE values are already included in Figure 6), (ii) group concurrent, lagged at total anomalies following the arrangement in Figure 6.

**References**

[revised manuscript text omitted]

**Formatted Table** ... [96]
**Formatted** ... [97]
**Formatted** ... [98]
**Formatted** ... [99]
**Formatted** ... [100]
**Formatted** ... [101]
**Formatted** ... [102]
**Formatted** ... [103]
**Formatted** ... [104]
**Formatted** ... [105]
**Formatted** ... [106]
**Formatted** ... [107]
**Formatted** ... [108]
**Formatted** ... [109]
**Formatted** ... [110]
**Formatted** ... [111]
**Formatted** ... [112]
**Formatted** ... [113]
**Formatted** ... [114]
**Formatted** ... [115]
**Formatted** ... [116]
**Formatted** ... [117]
**Formatted** ... [118]
**Formatted** ... [119]
**Formatted** ... [120]

**Formatted** ... [152]
**Formatted** ... [151]
**Formatted** ... [153]
**Formatted** ... [155]
**Formatted** ... [154]
**Formatted** ... [156]
**Formatted** ... [157]
**Formatted** ... [158]
**Formatted** ... [159]
**Formatted** ... [160]
**Formatted** ... [161]
**Formatted** ... [162]
**Formatted** ... [163]
**Formatted** ... [164]
**Formatted** ... [165]
**Formatted** ... [166]
**Formatted** ... [167]
**Formatted** ... [168]
**Formatted** ... [169]
**Formatted** ... [170]
**Formatted** ... [171]
**Formatted** ... [172]
**Formatted** ... [173]
**Formatted** ... [174]
**Formatted** ... [175]
**Formatted** ... [176]
**Formatted** ... [177]
**Formatted** ... [178]

**Table 5.** IAV of 2001-2015 regional and pan-tropical NBE lagged effects attributable to annual anomalies in column-denoted ecosystem states (Eq. 22), as % of total NBE lagged effects ($\Delta NBE^{LAG}$) IAV; bracketed values correspond to Pearson's correlation coefficients between single-state NBE lagged effects and total $\Delta NBE^{LAG}$; "*" denotes p-values < 0.05. The values highlighted in red denote the maximum contribution in each region.

| | Labile C | Foliar C | Fine Root C | Wood C | Litter C | Soil C | Plant-av. H$_2$O |
|---|---|---|---|---|---|---|---|
| SH South America | 9%(0.88*) | 48%(0.69*) | 15%(0.12) | 2%(-0.80*) | 27%(0.43) | 41%(0.85*) | 30%(0.28) |
| NH South America | 3%(0.88*) | 98%(0.94*) | 6%(0.48) | 6%(-0.91*) | 12%(0.17) | 34%(-0.17) | 28%(0.45) |
| Southern Africa | 6%(0.17) | 41%(0.69*) | 3%(0.66*) | 1%(0.58*) | 15%(0.85*) | 40%(0.78*) | 17%(0.85*) |
| Northern SS Africa | 35%(0.45) | 120%(0.64*) | 2%(-0.01) | 4%(-0.16) | 12%(-0.03) | 125%(-0.22) | 50%(0.58*) |
| Australia | 8%(0.71*) | 58%(0.68*) | 3%(-0.61*) | 1%(-0.53*) | 11%(-0.02) | 10%(0.17) | 54%(0.88*) |
| SE Asia & Indonesia | 7%(0.14) | 43%(0.16) | 18%(-0.63*) | 5%(0.29) | 35%(0.07) | 62%(0.45) | 64%(0.94*) |
| Wet Tropics | 8%(0.66*) | 99%(0.84*) | 14%(0.18) | 8%(-0.73*) | 27%(0.12) | 56%(-0.55*) | 37%(0.79*) |
| Dry Tropics | 16%(0.71*) | 47%(0.70*) | 6%(-0.09) | 1%(0.37) | 17%(0.38) | 13%(0.58*) | 43%(0.83*) |
| Tropics | 12%(0.82*) | 58%(0.83*) | 10%(0.03) | 3%(-0.51) | 21%(0.23) | 20%(-0.26) | 39%(0.82*) |

| Page 5: [1] Deleted | Anthony Bloom | 7/21/20 2:58:00 PM |
|---|---|---|

| Page 17: [2] Deleted | Anthony Bloom | 7/20/20 12:03:00 PM |
|---|---|---|

| Page 17: [2] Deleted | Anthony Bloom | 7/20/20 12:03:00 PM |
|---|---|---|

| Page 17: [2] Deleted | Anthony Bloom | 7/20/20 12:03:00 PM |
|---|---|---|

| Page 17: [2] Deleted | Anthony Bloom | 7/20/20 12:03:00 PM |
|---|---|---|

| Page 17: [2] Deleted | Anthony Bloom | 7/20/20 12:03:00 PM |
|---|---|---|

| Page 17: [2] Deleted | Anthony Bloom | 7/20/20 12:03:00 PM |
|---|---|---|

| Page 17: [2] Deleted | Anthony Bloom | 7/20/20 12:03:00 PM |
|---|---|---|

| Page 17: [3] Deleted | Anthony Bloom | 7/31/20 1:34:00 PM |
|---|---|---|

| Page 17: [3] Deleted | Anthony Bloom | 7/31/20 1:34:00 PM |
|---|---|---|

| Page 17: [3] Deleted | Anthony Bloom | 7/31/20 1:34:00 PM |
|---|---|---|

| Page 17: [3] Deleted | Anthony Bloom | 7/31/20 1:34:00 PM |
|---|---|---|

| Page 17: [3] Deleted | Anthony Bloom | 7/31/20 1:34:00 PM |
|---|---|---|

| Page 17: [3] Deleted | Anthony Bloom | 7/31/20 1:34:00 PM |
|---|---|---|

| Page 17: [3] Deleted | Anthony Bloom | 7/31/20 1:34:00 PM |
|---|---|---|

| Page 17: [3] Deleted | Anthony Bloom | 7/31/20 1:34:00 PM |
|---|---|---|

| | | |
|---|---|---|
| **Page 17: [3] Deleted** | **Anthony Bloom** | **7/31/20 1:34:00 PM** |
| **Page 17: [3] Deleted** | **Anthony Bloom** | **7/31/20 1:34:00 PM** |
| **Page 17: [3] Deleted** | **Anthony Bloom** | **7/31/20 1:34:00 PM** |
| **Page 17: [3] Deleted** | **Anthony Bloom** | **7/31/20 1:34:00 PM** |
| **Page 17: [3] Deleted** | **Anthony Bloom** | **7/31/20 1:34:00 PM** |
| **Page 17: [3] Deleted** | **Anthony Bloom** | **7/31/20 1:34:00 PM** |
| **Page 17: [3] Deleted** | **Anthony Bloom** | **7/31/20 1:34:00 PM** |
| **Page 17: [3] Deleted** | **Anthony Bloom** | **7/31/20 1:34:00 PM** |
| **Page 17: [3] Deleted** | **Anthony Bloom** | **7/31/20 1:34:00 PM** |
| **Page 17: [3] Deleted** | **Anthony Bloom** | **7/31/20 1:34:00 PM** |
| **Page 17: [3] Deleted** | **Anthony Bloom** | **7/31/20 1:34:00 PM** |
| **Page 17: [4] Deleted** | **Anthony Bloom** | **7/18/20 5:48:00 PM** |
| **Page 17: [4] Deleted** | **Anthony Bloom** | **7/18/20 5:48:00 PM** |
| **Page 17: [4] Deleted** | **Anthony Bloom** | **7/18/20 5:48:00 PM** |
| **Page 17: [4] Deleted** | **Anthony Bloom** | **7/18/20 5:48:00 PM** |
| **Page 17: [4] Deleted** | **Anthony Bloom** | **7/18/20 5:48:00 PM** |

| Page 17: [4] Deleted | Anthony Bloom | 7/18/20 5:48:00 PM |
| Page 17: [4] Deleted | Anthony Bloom | 7/18/20 5:48:00 PM |
| Page 17: [4] Deleted | Anthony Bloom | 7/18/20 5:48:00 PM |
| Page 17: [4] Deleted | Anthony Bloom | 7/18/20 5:48:00 PM |
| Page 17: [4] Deleted | Anthony Bloom | 7/18/20 5:48:00 PM |
| Page 17: [4] Deleted | Anthony Bloom | 7/18/20 5:48:00 PM |
| Page 17: [4] Deleted | Anthony Bloom | 7/18/20 5:48:00 PM |
| Page 17: [4] Deleted | Anthony Bloom | 7/18/20 5:48:00 PM |
| Page 17: [4] Deleted | Anthony Bloom | 7/18/20 5:48:00 PM |
| Page 17: [4] Deleted | Anthony Bloom | 7/18/20 5:48:00 PM |
| Page 17: [5] Deleted | Anthony Bloom | 7/20/20 3:30:00 PM |
| Page 17: [5] Deleted | Anthony Bloom | 7/20/20 3:30:00 PM |
| Page 17: [5] Deleted | Anthony Bloom | 7/20/20 3:30:00 PM |
| Page 17: [5] Deleted | Anthony Bloom | 7/20/20 3:30:00 PM |
| Page 17: [5] Deleted | Anthony Bloom | 7/20/20 3:30:00 PM |

| Page 17: [6] Formatted | Anthony Bloom | 7/31/20 2:24:00 PM |

Superscript

| Page 20: [7] Deleted | Anthony Bloom | 7/23/20 5:22:00 PM |

| Page 43: [8] Deleted | Anthony Bloom | 6/14/20 3:43:00 PM |

| Page 43: [8] Deleted | Anthony Bloom | 6/14/20 3:43:00 PM |

| Page 43: [8] Deleted | Anthony Bloom | 6/14/20 3:43:00 PM |

| Page 43: [9] Formatted | Anthony Bloom | 5/16/20 10:33:00 PM |

Subscript

| Page 43: [9] Formatted | Anthony Bloom | 5/16/20 10:33:00 PM |

Subscript

| Page 43: [10] Deleted | Anthony Bloom | 6/14/20 3:43:00 PM |

| Page 46: [11] Deleted | Anthony Bloom | 6/20/20 4:23:00 PM |

| Page 52: [12] Formatted | Anthony Bloom | 7/17/20 1:49:00 PM |

Font: 8 pt

| Page 52: [13] Formatted Table | Anthony Bloom | 7/17/20 1:49:00 PM |

Formatted Table

| Page 52: [14] Formatted | Anthony Bloom | 7/17/20 1:49:00 PM |

Font: 8 pt

| Page 52: [15] Formatted | Anthony Bloom | 7/17/20 1:49:00 PM |

Font: 8 pt

| Page 52: [16] Formatted | Anthony Bloom | 7/17/20 1:49:00 PM |

Font: 8 pt

| Page 52: [17] Formatted | Anthony Bloom | 7/17/20 1:49:00 PM |

Font: 8 pt

| Page 52: [18] Formatted | Anthony Bloom | 7/17/20 1:49:00 PM |

Font: 8 pt

| Page 52: [19] Formatted | Anthony Bloom | 7/17/20 1:49:00 PM |

**Page 52: [20] Formatted**        Anthony Bloom        7/17/20 1:49:00 PM

Font: 8 pt

**Page 52: [21] Formatted**        Anthony Bloom        7/17/20 1:49:00 PM

Font: 8 pt

**Page 52: [22] Formatted**        Anthony Bloom        7/17/20 1:49:00 PM

Font: 8 pt

**Page 52: [23] Formatted**        Anthony Bloom        7/17/20 1:49:00 PM

Font: 8 pt

**Page 52: [24] Formatted**        Anthony Bloom        7/17/20 1:49:00 PM

Font: 8 pt

**Page 52: [25] Formatted**        Anthony Bloom        7/17/20 1:49:00 PM

Font: 8 pt

**Page 52: [26] Formatted**        Anthony Bloom        7/17/20 1:49:00 PM

Font: 8 pt

**Page 52: [27] Formatted**        Anthony Bloom        7/17/20 1:49:00 PM

Font: 8 pt

**Page 52: [28] Formatted**        Anthony Bloom        7/17/20 1:49:00 PM

Font: 8 pt

**Page 52: [29] Formatted**        Anthony Bloom        7/17/20 1:49:00 PM

Font: 8 pt

**Page 52: [30] Formatted**        Anthony Bloom        7/17/20 1:49:00 PM

Font: 8 pt

**Page 52: [31] Formatted**        Anthony Bloom        7/17/20 1:49:00 PM

Font: 8 pt

**Page 52: [32] Formatted**        Anthony Bloom        7/17/20 1:49:00 PM

Font: 8 pt

**Page 52: [33] Formatted**        Anthony Bloom        7/17/20 1:49:00 PM

Font: 8 pt

**Page 52: [34] Formatted**        Anthony Bloom        7/17/20 1:49:00 PM

Font: 8 pt

**Page 52: [35] Formatted**        Anthony Bloom        7/17/20 1:49:00 PM

Font: 8 pt

| Page 52: [37] Formatted | Anthony Bloom | 7/17/20 1:49:00 PM |
|---|---|---|

Font: 8 pt

| Page 52: [38] Formatted | Anthony Bloom | 7/17/20 1:49:00 PM |
|---|---|---|

Font: 8 pt

| Page 52: [39] Formatted | Anthony Bloom | 7/17/20 1:49:00 PM |
|---|---|---|

Font: 8 pt

| Page 52: [40] Formatted | Anthony Bloom | 7/17/20 1:49:00 PM |
|---|---|---|

Font: 8 pt

| Page 52: [41] Formatted | Anthony Bloom | 7/17/20 1:49:00 PM |
|---|---|---|

Font: 8 pt

| Page 52: [42] Formatted | Anthony Bloom | 7/17/20 1:49:00 PM |
|---|---|---|

Font: 8 pt

| Page 52: [43] Formatted | Anthony Bloom | 7/17/20 1:49:00 PM |
|---|---|---|

Font: 8 pt

| Page 52: [44] Formatted | Anthony Bloom | 7/17/20 1:49:00 PM |
|---|---|---|

Font: 8 pt

| Page 52: [45] Formatted | Anthony Bloom | 7/17/20 1:49:00 PM |
|---|---|---|

Font: 8 pt

| Page 52: [46] Formatted | Anthony Bloom | 7/17/20 1:49:00 PM |
|---|---|---|

Font: 8 pt

| Page 52: [47] Formatted | Anthony Bloom | 7/17/20 1:49:00 PM |
|---|---|---|

Font: 8 pt

| Page 52: [48] Formatted | Anthony Bloom | 7/17/20 1:49:00 PM |
|---|---|---|

Font: 8 pt

| Page 52: [49] Formatted | Anthony Bloom | 7/17/20 1:49:00 PM |
|---|---|---|

Font: 8 pt

| Page 52: [50] Formatted | Anthony Bloom | 7/17/20 1:49:00 PM |
|---|---|---|

Font: 8 pt

| Page 52: [51] Formatted | Anthony Bloom | 7/17/20 1:49:00 PM |
|---|---|---|

Font: 8 pt

| Page 52: [52] Formatted | Anthony Bloom | 7/17/20 1:49:00 PM |
|---|---|---|

Font: 8 pt

Font: 8 pt

| Page 52: [54] Formatted | Anthony Bloom | 7/17/20 1:49:00 PM |

Font: 8 pt

| Page 52: [55] Formatted | Anthony Bloom | 7/17/20 1:49:00 PM |

Font: 8 pt

| Page 52: [56] Formatted | Anthony Bloom | 7/17/20 1:49:00 PM |

Font: 8 pt

| Page 52: [57] Formatted | Anthony Bloom | 7/17/20 1:49:00 PM |

Font: 8 pt

| Page 53: [58] Deleted | Anthony Bloom | 8/1/20 5:32:00 PM |

| Page 53: [58] Deleted | Anthony Bloom | 8/1/20 5:32:00 PM |

| Page 53: [59] Formatted Table | Anthony Bloom | 7/4/20 5:55:00 PM |

Formatted Table

| Page 53: [60] Formatted | Anthony Bloom | 7/4/20 5:55:00 PM |

Font: 8 pt

| Page 53: [61] Formatted | Anthony Bloom | 7/4/20 5:55:00 PM |

Font: 8 pt

| Page 53: [62] Formatted | Anthony Bloom | 7/4/20 5:55:00 PM |

Font: 8 pt

| Page 53: [63] Formatted | Anthony Bloom | 7/4/20 5:55:00 PM |

Font: 8 pt

| Page 53: [64] Formatted | Anthony Bloom | 7/4/20 5:55:00 PM |

Font: 8 pt

| Page 53: [65] Formatted | Anthony Bloom | 7/4/20 5:55:00 PM |

Font: 8 pt

| Page 53: [66] Formatted | Anthony Bloom | 7/4/20 5:55:00 PM |

Font: 8 pt

| Page 53: [67] Formatted | Anthony Bloom | 7/4/20 5:55:00 PM |

Font: 8 pt

| Page 53: [69] Formatted | Anthony Bloom | 7/4/20 5:55:00 PM |
|---|---|---|

Font: 8 pt

| Page 53: [70] Formatted | Anthony Bloom | 7/4/20 5:55:00 PM |
|---|---|---|

Font: 8 pt

| Page 53: [71] Formatted | Anthony Bloom | 7/4/20 5:55:00 PM |
|---|---|---|

Font: 8 pt

| Page 53: [72] Formatted | Anthony Bloom | 7/4/20 5:55:00 PM |
|---|---|---|

Font: 8 pt

| Page 53: [73] Formatted | Anthony Bloom | 7/4/20 5:55:00 PM |
|---|---|---|

Font: 8 pt

| Page 53: [74] Formatted | Anthony Bloom | 7/4/20 5:55:00 PM |
|---|---|---|

Font: 8 pt

| Page 53: [75] Formatted | Anthony Bloom | 7/4/20 5:55:00 PM |
|---|---|---|

Font: 8 pt

| Page 53: [76] Formatted | Anthony Bloom | 7/4/20 5:55:00 PM |
|---|---|---|

Font: 8 pt

| Page 53: [77] Formatted | Anthony Bloom | 7/4/20 5:55:00 PM |
|---|---|---|

Font: 8 pt

| Page 53: [78] Formatted | Anthony Bloom | 7/4/20 5:55:00 PM |
|---|---|---|

Font: 8 pt

| Page 53: [79] Formatted | Anthony Bloom | 7/4/20 5:55:00 PM |
|---|---|---|

Font: 8 pt

| Page 53: [80] Formatted | Anthony Bloom | 7/4/20 5:55:00 PM |
|---|---|---|

Font: 8 pt

| Page 53: [81] Formatted | Anthony Bloom | 7/4/20 5:55:00 PM |
|---|---|---|

Font: 8 pt

| Page 53: [82] Formatted | Anthony Bloom | 7/4/20 5:55:00 PM |
|---|---|---|

Font: 8 pt

| Page 53: [83] Formatted | Anthony Bloom | 7/4/20 5:55:00 PM |
|---|---|---|

Font: 8 pt

| Page 53: [84] Formatted | Anthony Bloom | 7/4/20 5:55:00 PM |
|---|---|---|

Font: 8 pt

| Page 53: [86] Formatted | Anthony Bloom | 7/4/20 5:55:00 PM |

Font: 8 pt

| Page 53: [87] Formatted | Anthony Bloom | 7/4/20 5:55:00 PM |

Font: 8 pt

| Page 53: [88] Formatted | Anthony Bloom | 7/4/20 5:55:00 PM |

Font: 8 pt

| Page 53: [89] Formatted | Anthony Bloom | 7/4/20 5:55:00 PM |

Font: 8 pt

| Page 53: [90] Formatted | Anthony Bloom | 7/4/20 5:55:00 PM |

Font: 8 pt

| Page 53: [91] Formatted | Anthony Bloom | 7/4/20 5:55:00 PM |

Font: 8 pt

| Page 53: [92] Formatted | Anthony Bloom | 7/4/20 5:55:00 PM |

Font: 8 pt

| Page 53: [93] Formatted | Anthony Bloom | 7/4/20 5:55:00 PM |

Font: 8 pt

| Page 53: [94] Formatted | Anthony Bloom | 7/4/20 5:55:00 PM |

Font: 8 pt

| Page 53: [95] Formatted | Anthony Bloom | 7/4/20 5:55:00 PM |

Font: 8 pt

| Page 54: [96] Formatted Table | Anthony Bloom | 7/4/20 9:41:00 PM |

Formatted Table

| Page 54: [97] Formatted | Anthony Bloom | 7/4/20 9:46:00 PM |

Font: (Default) Times New Roman, 8 pt, Bold, Font color: Red

| Page 54: [98] Formatted | Anthony Bloom | 7/4/20 9:40:00 PM |

Font: (Default) Times New Roman, 8 pt

| Page 54: [99] Formatted | Anthony Bloom | 7/4/20 9:40:00 PM |

Font: (Default) Times New Roman, 8 pt

| Page 54: [100] Formatted | Anthony Bloom | 7/4/20 9:47:00 PM |

Font: (Default) Times New Roman, 8 pt, Bold, Font color: Red

| Page 54: [101] Formatted | Anthony Bloom | 7/4/20 9:40:00 PM |

Font: (Default) Times New Roman, 8 pt

Font: (Default) Times New Roman, 8 pt

| Page 54: [103] Formatted | Anthony Bloom | 7/4/20 9:40:00 PM |
|---|---|---|

Font: (Default) Times New Roman, 8 pt

| Page 54: [103] Formatted | Anthony Bloom | 7/4/20 9:40:00 PM |
|---|---|---|

Font: (Default) Times New Roman, 8 pt

| Page 54: [104] Formatted | Anthony Bloom | 7/4/20 9:40:00 PM |
|---|---|---|

Font: (Default) Times New Roman, 8 pt

| Page 54: [105] Formatted | Anthony Bloom | 7/4/20 9:40:00 PM |
|---|---|---|

Font: (Default) Times New Roman, 8 pt

| Page 54: [106] Formatted | Anthony Bloom | 7/4/20 9:47:00 PM |
|---|---|---|

Font: (Default) Times New Roman, 8 pt, Bold, Font color: Red

| Page 54: [107] Formatted | Anthony Bloom | 7/4/20 9:40:00 PM |
|---|---|---|

Font: (Default) Times New Roman, 8 pt

| Page 54: [108] Formatted | Anthony Bloom | 7/4/20 9:40:00 PM |
|---|---|---|

Font: (Default) Times New Roman, 8 pt

| Page 54: [109] Formatted | Anthony Bloom | 7/4/20 9:40:00 PM |
|---|---|---|

Font: (Default) Times New Roman, 8 pt

| Page 54: [109] Formatted | Anthony Bloom | 7/4/20 9:40:00 PM |
|---|---|---|

Font: (Default) Times New Roman, 8 pt

| Page 54: [110] Formatted | Anthony Bloom | 7/4/20 9:40:00 PM |
|---|---|---|

Font: (Default) Times New Roman, 8 pt

| Page 54: [111] Formatted | Anthony Bloom | 7/4/20 9:40:00 PM |
|---|---|---|

Font: (Default) Times New Roman, 8 pt

| Page 54: [112] Formatted | Anthony Bloom | 7/4/20 9:48:00 PM |
|---|---|---|

Font: (Default) Times New Roman, 8 pt, Bold, Font color: Red

| Page 54: [113] Formatted | Anthony Bloom | 7/4/20 9:40:00 PM |
|---|---|---|

Font: (Default) Times New Roman, 8 pt

| Page 54: [114] Formatted | Anthony Bloom | 7/4/20 9:40:00 PM |
|---|---|---|

Font: (Default) Times New Roman, 8 pt

| Page 54: [115] Formatted | Anthony Bloom | 7/4/20 9:40:00 PM |
|---|---|---|

Font: (Default) Times New Roman, 8 pt

**Page 54: [116] Formatted**         **Anthony Bloom**         **7/4/20 9:40:00 PM**

Font: (Default) Times New Roman, 8 pt

**Page 54: [117] Formatted**         **Anthony Bloom**         **7/4/20 9:40:00 PM**

Font: (Default) Times New Roman, 8 pt

**Page 54: [118] Formatted**         **Anthony Bloom**         **7/4/20 9:40:00 PM**

Font: (Default) Times New Roman, 8 pt

**Page 54: [118] Formatted**         **Anthony Bloom**         **7/4/20 9:40:00 PM**

Font: (Default) Times New Roman, 8 pt

**Page 54: [119] Formatted**         **Anthony Bloom**         **7/4/20 9:40:00 PM**

Font: (Default) Times New Roman, 8 pt

**Page 54: [120] Formatted**         **Anthony Bloom**         **7/4/20 9:40:00 PM**

Font: (Default) Times New Roman, 8 pt

**Page 54: [121] Formatted**         **Anthony Bloom**         **7/4/20 9:47:00 PM**

Font: (Default) Times New Roman, 8 pt, Bold, Font color: Red

**Page 54: [122] Formatted**         **Anthony Bloom**         **7/4/20 9:40:00 PM**

Font: (Default) Times New Roman, 8 pt

**Page 54: [123] Formatted**         **Anthony Bloom**         **7/4/20 9:40:00 PM**

Font: (Default) Times New Roman, 8 pt

**Page 54: [124] Formatted**         **Anthony Bloom**         **7/4/20 9:48:00 PM**

Font: (Default) Times New Roman, 8 pt, Bold, Font color: Red

**Page 54: [125] Formatted**         **Anthony Bloom**         **7/4/20 9:40:00 PM**

Font: (Default) Times New Roman, 8 pt

**Page 54: [126] Formatted**         **Anthony Bloom**         **7/4/20 9:40:00 PM**

Font: (Default) Times New Roman, 8 pt

**Page 54: [127] Formatted**         **Anthony Bloom**         **7/4/20 9:47:00 PM**

Font: (Default) Times New Roman, 8 pt, Bold, Font color: Red

**Page 54: [128] Formatted**         **Anthony Bloom**         **7/4/20 9:40:00 PM**

Font: (Default) Times New Roman, 8 pt

**Page 54: [129] Formatted**         **Anthony Bloom**         **7/4/20 9:40:00 PM**

Font: (Default) Times New Roman, 8 pt

**Page 54: [130] Formatted**         **Anthony Bloom**         **7/4/20 9:48:00 PM**

Font: (Default) Times New Roman, 8 pt, Bold, Font color: Red

**Page 54: [132] Formatted**         Anthony Bloom         7/4/20 9:40:00 PM

Font: (Default) Times New Roman, 8 pt

**Page 54: [133] Formatted**         Anthony Bloom         7/4/20 9:47:00 PM

Font: (Default) Times New Roman, 8 pt, Bold, Font color: Red

**Page 54: [134] Formatted**         Anthony Bloom         7/4/20 9:40:00 PM

Font: (Default) Times New Roman, 8 pt

**Page 54: [135] Formatted**         Anthony Bloom         7/4/20 9:40:00 PM

Font: (Default) Times New Roman, 8 pt

**Page 54: [136] Formatted**         Anthony Bloom         7/4/20 9:48:00 PM

Font: (Default) Times New Roman, 8 pt, Bold, Font color: Red

**Page 54: [137] Formatted**         Anthony Bloom         7/4/20 9:40:00 PM

Font: (Default) Times New Roman, 8 pt

**Page 54: [138] Formatted**         Anthony Bloom         7/4/20 9:40:00 PM

Font: (Default) Times New Roman, 8 pt

**Page 54: [139] Formatted**         Anthony Bloom         7/4/20 9:47:00 PM

Font: (Default) Times New Roman, 8 pt, Bold, Font color: Red

**Page 54: [140] Formatted**         Anthony Bloom         7/4/20 9:40:00 PM

Font: (Default) Times New Roman, 8 pt

**Page 54: [141] Formatted**         Anthony Bloom         7/4/20 9:40:00 PM

Font: (Default) Times New Roman, 8 pt

**Page 54: [142] Formatted**         Anthony Bloom         7/4/20 9:48:00 PM

Font: (Default) Times New Roman, 8 pt, Bold, Font color: Red

**Page 54: [143] Formatted**         Anthony Bloom         7/4/20 9:40:00 PM

Font: (Default) Times New Roman, 8 pt

**Page 54: [144] Formatted**         Anthony Bloom         7/4/20 9:40:00 PM

Font: (Default) Times New Roman, 8 pt

**Page 54: [145] Formatted**         Anthony Bloom         7/4/20 9:47:00 PM

Font: (Default) Times New Roman, 8 pt, Bold, Font color: Red

**Page 54: [146] Formatted**         Anthony Bloom         7/4/20 9:40:00 PM

Font: (Default) Times New Roman, 8 pt

**Page 54: [147] Formatted**         Anthony Bloom         7/4/20 9:40:00 PM

Font: (Default) Times New Roman, 8 pt

Font: (Default) Times New Roman, 8 pt

| Page 54: [148] Formatted | Anthony Bloom | 7/4/20 9:40:00 PM |
|---|---|---|

Font: (Default) Times New Roman, 8 pt

| Page 54: [149] Formatted | Anthony Bloom | 7/4/20 9:40:00 PM |
|---|---|---|

Font: (Default) Times New Roman, 8 pt

| Page 54: [150] Formatted | Anthony Bloom | 7/4/20 9:40:00 PM |
|---|---|---|

Font: (Default) Times New Roman, 8 pt

| Page 55: [151] Formatted | Anthony Bloom | 7/4/20 9:44:00 PM |
|---|---|---|

Font: (Default) Times New Roman, 8 pt

| Page 55: [152] Formatted | Anthony Bloom | 7/22/20 5:03:00 PM |
|---|---|---|

Font: (Default) Times New Roman, 8 pt, Font color: Red

| Page 55: [152] Formatted | Anthony Bloom | 7/22/20 5:03:00 PM |
|---|---|---|

Font: (Default) Times New Roman, 8 pt, Font color: Red

| Page 55: [153] Formatted | Anthony Bloom | 7/22/20 5:03:00 PM |
|---|---|---|

Font: Bold, Font color: Red

| Page 55: [154] Formatted | Anthony Bloom | 7/4/20 9:44:00 PM |
|---|---|---|

Font: (Default) Times New Roman, 8 pt

| Page 55: [155] Formatted | Anthony Bloom | 7/4/20 9:44:00 PM |
|---|---|---|

Font: (Default) Times New Roman, 8 pt

| Page 55: [156] Formatted | Anthony Bloom | 7/4/20 9:44:00 PM |
|---|---|---|

Font: (Default) Times New Roman, 8 pt

| Page 55: [157] Formatted | Anthony Bloom | 7/4/20 9:44:00 PM |
|---|---|---|

Font: (Default) Times New Roman, 8 pt

| Page 55: [158] Formatted | Anthony Bloom | 7/4/20 9:44:00 PM |
|---|---|---|

Font: (Default) Times New Roman, 8 pt

| Page 55: [159] Formatted | Anthony Bloom | 7/4/20 9:44:00 PM |
|---|---|---|

Font: (Default) Times New Roman, 8 pt

| Page 55: [160] Formatted | Anthony Bloom | 7/22/20 5:03:00 PM |
|---|---|---|

Font: (Default) Times New Roman, 8 pt, Font color: Red

| Page 55: [160] Formatted | Anthony Bloom | 7/22/20 5:03:00 PM |
|---|---|---|

Font: (Default) Times New Roman, 8 pt, Font color: Red

| Page 55: [162] Formatted | Anthony Bloom | 7/4/20 9:44:00 PM |

Font: (Default) Times New Roman, 8 pt

| Page 55: [163] Formatted | Anthony Bloom | 7/4/20 9:44:00 PM |

Font: (Default) Times New Roman, 8 pt

| Page 55: [164] Formatted | Anthony Bloom | 7/4/20 9:44:00 PM |

Font: (Default) Times New Roman, 8 pt

| Page 55: [165] Formatted | Anthony Bloom | 7/4/20 9:44:00 PM |

Font: (Default) Times New Roman, 8 pt

| Page 55: [166] Formatted | Anthony Bloom | 7/4/20 9:44:00 PM |

Font: (Default) Times New Roman, 8 pt

| Page 55: [167] Formatted | Anthony Bloom | 7/4/20 9:44:00 PM |

Font: (Default) Times New Roman, 8 pt

| Page 55: [168] Formatted | Anthony Bloom | 7/22/20 5:03:00 PM |

Font: (Default) Times New Roman, 8 pt, Font color: Red

| Page 55: [169] Formatted | Anthony Bloom | 7/4/20 9:44:00 PM |

Font: (Default) Times New Roman, 8 pt

| Page 55: [170] Formatted | Anthony Bloom | 7/4/20 9:44:00 PM |

Font: (Default) Times New Roman, 8 pt

| Page 55: [171] Formatted | Anthony Bloom | 7/4/20 9:44:00 PM |

Font: (Default) Times New Roman, 8 pt

| Page 55: [172] Formatted | Anthony Bloom | 7/4/20 9:44:00 PM |

Font: (Default) Times New Roman, 8 pt

| Page 55: [173] Formatted | Anthony Bloom | 7/4/20 9:44:00 PM |

Font: (Default) Times New Roman, 8 pt

| Page 55: [174] Formatted | Anthony Bloom | 7/4/20 9:44:00 PM |

Font: (Default) Times New Roman, 8 pt

| Page 55: [175] Formatted | Anthony Bloom | 7/4/20 9:44:00 PM |

Font: (Default) Times New Roman, 8 pt

| Page 55: [176] Formatted | Anthony Bloom | 7/4/20 9:44:00 PM |

Font: (Default) Times New Roman, 8 pt

| | | |
|---|---|---|
| **Page 55: [178] Formatted** | **Anthony Bloom** | **7/4/20 9:44:00 PM** |

Font: (Default) Times New Roman, 8 pt

| | | |
|---|---|---|
| **Page 55: [179] Formatted** | **Anthony Bloom** | **7/4/20 9:44:00 PM** |

Font: (Default) Times New Roman, 8 pt

| | | |
|---|---|---|
| **Page 55: [180] Formatted** | **Anthony Bloom** | **7/4/20 9:44:00 PM** |

Font: (Default) Times New Roman, 8 pt

| | | |
|---|---|---|
| **Page 55: [181] Formatted** | **Anthony Bloom** | **7/4/20 9:44:00 PM** |

Font: (Default) Times New Roman, 8 pt

| | | |
|---|---|---|
| **Page 55: [182] Formatted** | **Anthony Bloom** | **7/22/20 5:03:00 PM** |

Font: (Default) Times New Roman, 8 pt, Font color: Red

| | | |
|---|---|---|
| **Page 55: [183] Formatted** | **Anthony Bloom** | **7/4/20 9:44:00 PM** |

Font: (Default) Times New Roman, 8 pt

| | | |
|---|---|---|
| **Page 55: [184] Formatted** | **Anthony Bloom** | **7/4/20 9:44:00 PM** |

Font: (Default) Times New Roman, 8 pt

| | | |
|---|---|---|
| **Page 55: [185] Formatted** | **Anthony Bloom** | **7/4/20 9:44:00 PM** |

Font: (Default) Times New Roman, 8 pt

| | | |
|---|---|---|
| **Page 55: [186] Formatted** | **Anthony Bloom** | **7/4/20 9:44:00 PM** |

Font: (Default) Times New Roman, 8 pt

| | | |
|---|---|---|
| **Page 55: [187] Formatted** | **Anthony Bloom** | **7/4/20 9:44:00 PM** |

Font: (Default) Times New Roman, 8 pt

| | | |
|---|---|---|
| **Page 55: [188] Formatted** | **Anthony Bloom** | **7/4/20 9:44:00 PM** |

Font: (Default) Times New Roman, 8 pt

| | | |
|---|---|---|
| **Page 55: [189] Formatted** | **Anthony Bloom** | **7/4/20 9:44:00 PM** |

Font: (Default) Times New Roman, 8 pt

| | | |
|---|---|---|
| **Page 55: [190] Formatted** | **Anthony Bloom** | **7/4/20 9:44:00 PM** |

Font: (Default) Times New Roman, 8 pt

| | | |
|---|---|---|
| **Page 55: [191] Formatted** | **Anthony Bloom** | **7/4/20 9:44:00 PM** |

Font: (Default) Times New Roman, 8 pt

Font: (Default) Times New Roman, 8 pt

| Page 55: [193] Formatted | Anthony Bloom | 7/4/20 9:44:00 PM |
|---|---|---|

Font: (Default) Times New Roman, 8 pt

| Page 55: [194] Formatted | Anthony Bloom | 7/22/20 5:03:00 PM |
|---|---|---|

Font: (Default) Times New Roman, 8 pt, Bold, Font color: Red

| Page 55: [195] Formatted | Anthony Bloom | 7/4/20 9:44:00 PM |
|---|---|---|

Font: (Default) Times New Roman, 8 pt

| Page 55: [196] Formatted | Anthony Bloom | 7/22/20 5:03:00 PM |
|---|---|---|

Font: (Default) Times New Roman, 8 pt, Font color: Red

| Page 55: [196] Formatted | Anthony Bloom | 7/22/20 5:03:00 PM |
|---|---|---|

Font: (Default) Times New Roman, 8 pt, Font color: Red

| Page 55: [197] Formatted | Anthony Bloom | 7/22/20 5:03:00 PM |
|---|---|---|

Font: Bold, Font color: Red

| Page 55: [198] Formatted | Anthony Bloom | 7/4/20 9:44:00 PM |
|---|---|---|

Font: (Default) Times New Roman, 8 pt

| Page 55: [199] Formatted | Anthony Bloom | 7/4/20 9:44:00 PM |
|---|---|---|

Font: (Default) Times New Roman, 8 pt

| Page 55: [200] Formatted | Anthony Bloom | 7/4/20 9:44:00 PM |
|---|---|---|

Font: (Default) Times New Roman, 8 pt

| Page 55: [201] Formatted | Anthony Bloom | 7/4/20 9:44:00 PM |
|---|---|---|

Font: (Default) Times New Roman, 8 pt

| Page 55: [202] Formatted | Anthony Bloom | 7/4/20 9:44:00 PM |
|---|---|---|

Font: (Default) Times New Roman, 8 pt

| Page 55: [203] Formatted | Anthony Bloom | 7/4/20 9:44:00 PM |
|---|---|---|

Font: (Default) Times New Roman, 8 pt

| Page 55: [204] Formatted | Anthony Bloom | 7/22/20 5:03:00 PM |
|---|---|---|

Font: (Default) Times New Roman, 8 pt, Font color: Red

| Page 55: [204] Formatted | Anthony Bloom | 7/22/20 5:03:00 PM |
|---|---|---|

Font: (Default) Times New Roman, 8 pt, Font color: Red

| Page 55: [205] Formatted | Anthony Bloom | 7/4/20 9:44:00 PM |
|---|---|---|

Font: (Default) Times New Roman, 8 pt

| Page 55: [207] Formatted | Anthony Bloom | 7/4/20 9:44:00 PM |
|---|---|---|

Font: (Default) Times New Roman, 8 pt

| Page 55: [208] Formatted | Anthony Bloom | 7/4/20 9:44:00 PM |
|---|---|---|

Font: (Default) Times New Roman, 8 pt

| Page 55: [209] Formatted | Anthony Bloom | 7/4/20 9:44:00 PM |
|---|---|---|

Font: (Default) Times New Roman, 8 pt

| Page 55: [210] Formatted | Anthony Bloom | 7/4/20 9:44:00 PM |
|---|---|---|

Font: (Default) Times New Roman, 8 pt

| Page 55: [211] Formatted | Anthony Bloom | 7/22/20 5:03:00 PM |
|---|---|---|

Font: (Default) Times New Roman, 8 pt, Bold, Font color: Red

| Page 55: [212] Formatted | Anthony Bloom | 7/22/20 5:03:00 PM |
|---|---|---|

Font: Bold, Font color: Red

| Page 55: [213] Formatted | Anthony Bloom | 7/4/20 9:44:00 PM |
|---|---|---|

Font: (Default) Times New Roman, 8 pt

| Page 55: [214] Formatted | Anthony Bloom | 7/4/20 9:44:00 PM |
|---|---|---|

Font: (Default) Times New Roman, 8 pt

| Page 55: [215] Formatted | Anthony Bloom | 7/4/20 9:44:00 PM |
|---|---|---|

Font: (Default) Times New Roman, 8 pt

| Page 55: [216] Formatted | Anthony Bloom | 7/4/20 9:44:00 PM |
|---|---|---|

Font: (Default) Times New Roman, 8 pt

| Page 55: [217] Formatted | Anthony Bloom | 7/4/20 9:44:00 PM |
|---|---|---|

Font: (Default) Times New Roman, 8 pt